# 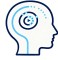 VideoLucy: Deep Memory Backtracking for Long Video Understanding

**Jialong Zuo**[1]   **Yongtai Deng**[1]   **Lingdong Kong**[2]   **Jingkang Yang**[3]   **Rui Jin**[1]
**Yiwei Zhang**[1]   **Nong Sang**[1]   **Liang Pan**[4]*   **Ziwei Liu**[3]   **Changxin Gao**[1]*

[1] National Key Laboratory of Multispectral Information Intelligent Processing Technology,
School of Artificial Intelligence and Automation, Huazhong University of Science and Technology
[2]NUS   [3]S-Lab, NTU   [4]Shanghai AI Lab

{jlongzuo, cgao}@hust.edu.cn

 **Code & Dataset:** videolucy.github.io

## Abstract

Recent studies have shown that agent-based systems leveraging large language models (LLMs) for key information retrieval and integration have emerged as a promising approach for long video understanding. However, these systems face two major challenges. First, they typically perform modeling and reasoning on individual frames, struggling to capture the temporal context of consecutive frames. Second, to reduce the cost of dense frame-level captioning, they adopt sparse frame sampling, which risks discarding crucial information. To overcome these limitations, we propose VideoLucy, a deep memory backtracking framework for long video understanding. Inspired by the human recollection process from coarse to fine, VideoLucy employs a *hierarchical memory structure* with progressive granularity. This structure explicitly defines the detail level and temporal scope of memory at different hierarchical depths. Through an *agent-based iterative backtracking* mechanism, VideoLucy systematically mines video-wide, question-relevant deep memories until sufficient information is gathered to provide a confident answer. This design enables effective temporal understanding of consecutive frames while preserving critical details. In addition, we introduce EgoMem, a new benchmark for long video understanding. EgoMem is designed to comprehensively evaluate a model's ability to understand complex events that unfold over time and capture fine-grained details in extremely long videos. Extensive experiments demonstrate the superiority of VideoLucy. Built on open-source models, VideoLucy significantly outperforms state-of-the-art methods on multiple long video understanding benchmarks, achieving performance even surpassing the latest proprietary models such as GPT-4o. Our code and dataset will be made publicly available.

## 1 Introduction

> *"Mom, I can feel my brain, the deepest parts of my memory. I can remember the feeling of your hand on my forehead when I ran a fever. I remember stroking the cat, it was so soft, a Siamese with blue eyes and a broken tail."*
>
> *—Lucy*

Long video understanding [67, 44, 11] is a highly concerned task, with the core objective of accurately and objectively answering various user questions based on the entire video content. This process demands that a system possess a comprehensive memory and grasp of almost all details within the video; otherwise, information gaps could lead to inaccurate answers.

---

*Corresponding Authors.

39th Conference on Neural Information Processing Systems (NeurIPS 2025).

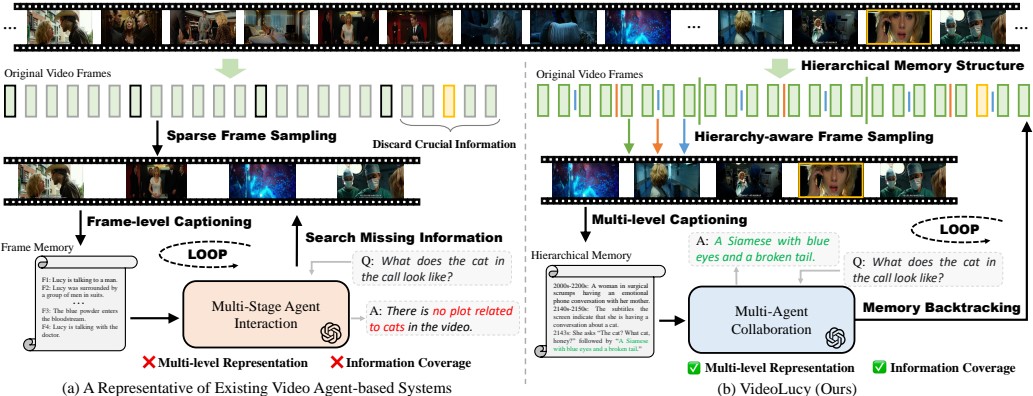

Figure 1: Comparison between our VideoLucy with existing video agent-based systems [32, 47]. In (a), they usually perform frame-level captioning on sparsely sampled frames, and then search for information, resulting in great information loss and hampering temporal understanding. In (b), our VideoLucy, through a hierarchical memory structure and a memory backtracking mechanism, effectively performs multi-level video representation and achieves comprehensive information coverage.

This need for comprehensive memory brings to mind a scene from the movie *Lucy*. The protagonist, Lucy, gains full access to her brain's potential due to an accident, acquiring an exceptionally powerful memory. She can recall every detail of her life from birth, even the sensation of her mother stroking her forehead during infancy. This extraordinary ability to trace back and precisely capture all information, whether instantaneous frames or continuous events, is exactly the goal we pursue in the long video understanding task.

Recently, the agent-based systems [9, 48, 32, 59] have emerged as a promising approach for long video understanding. In contrast to traditional video multi-modal large language models (MLLMs) [58, 37, 63], which have difficulty handling extremely long visual inputs, the agent-based systems typically harness the reasoning, planning, and memory capacities of large language models (LLMs) [1]. They iteratively search for and consolidate key information related to the question, thereby facilitating a more proficient understanding of long videos. Nevertheless, owing to the ineffective modeling of comprehensive memory, these systems still confront two significant challenges.

On the one hand, they typically perform modeling and reasoning on individual frames, struggling to capture the temporal context of consecutive frames. Essentially, they utilize a pre-trained caption model [64] to generate text descriptions for each specified frame in the video. Then, taking LLMs [1, 15] as the core, they construct an iterative information search loop to obtain the question-relevant key frames and their supplementary descriptions. For example, DrVideo [32] retrieves a set of key frames through the initial video frame-level documents. Then, it utilizes a multi-stage agent interaction loop to gradually update the text information of this set. However, in real-world applications of long video understanding, many user questions are closely linked to the temporal context of consecutive frames. These systems, which rely on the information of individual frames in isolation, have relatively weak capabilities in temporal understanding.

On the other hand, they usually adopt sparse frame sampling to reduce the cost of dense frame-level captioning, which risks discarding crucial information. Obviously, generating captions for each frame of a long video demands vast computational resources and much time. For instance, even if we generate captions for each frame of a one-hour video (with 1 FPS), a total of 3,600 captions need to be produced and processed subsequently. Therefore, these systems adopt a compromising measure, namely sparse frame sampling. For example, VideoTree [48] preprocesses videos by sampling the original frames at 0.125 FPS for Video-MME [11]. Explicitly, this sparse sampling compromise will lead to the loss of much crucial detailed information for long video understanding.

To address these limitations above, as shown in Fig. 1, we propose VideoLucy, a deep memory backtracking framework for long video understanding. Drawing from cognitive science, human recollection typically proceeds from coarse to fine, starting with general impressions and gradually retrieving finer details. VideoLucy also employs a *hierarchical memory structure* with progressive granularity. This structure explicitly defines the detail level and temporal scope of memory at different hierarchical depths. As the memory delves deeper, the temporal scope shrinks dynamically while the detail level increases gradually. Additionally, with an *agent-based iterative backtracking*

mechanism proposed, VideoLucy systematically mines video-wide, question-relevant deep memories until sufficient information is gathered to provide a confident answer. In essence, just as humans move from a fuzzy to a sharp memory when recalling the past, VideoLucy also starts with a vague memory of the entire video and gradually delves into the detailed memories relevant to the questions. This design enables VideoLucy to recall a comprehensive memory of the entire video, achieving effective temporal understanding while retaining crucial information. Just like Lucy in the movie, VideoLucy can also say, "I can feel my brain, the deepest parts of my memory."

In addition, we introduce EgoMem, a new benchmark for long video understanding. Built on EgoLife [51], EgoMem comprehensively assesses a model's temporal understanding and fine-grained detail perception in extremely long videos. Through six question-answering designs, it conducts multi-dimensional evaluations on the model's comprehension of complex, time-evolving events in first-person daily life recordings. Additionally, EgoMem gauges the model's capacity to detect fleeting detailed visual features in long videos, such as those that manifest for merely a few seconds. It contains 42 videos and 504 QAs in total, with each video averaging about 6.33 hours in length.

We conduct extensive experiments to verify the superiority of our VideoLucy. Built on open-source models [2, 15], VideoLucy significantly outperforms state-of-the-art methods on multiple long video understanding benchmarks [67, 11, 44], achieving performance even surpassing the latest proprietary models such as GPT-4o [1]. For example, on LVBench [44], our VideoLucy with Qwen2.5-VL-7B [2] as the captioner achieves an accuracy of 58.8%, representing an 9.9% improvement compared to GPT-4o. Ablation studies and analyses further validate the effectiveness of the hierarchical memory structure, and the "needle in a video haystack" experiment showcases its unprecedented detail perception capability. As an agent system with dynamic and comprehensive memory, our VideoLucy can pave the way for future research in this rapidly evolving field of long video understanding.

## 2 Method

We present VideoLucy, a deep memory backtracking framework for long video understanding, which dynamically recalls comprehensive and in-depth memories of the entire video based on the question, so as to achieve accurate answers. First, we propose a hierarchical memory structure that conforms to the pattern of transitioning from coarse to fine in the human recollection process, thus enabling the efficient modeling of abundant information in long videos. (Sec. 2.1). Next, we present the agents which are empowered with different roles through prompt engineering, and these agents accomplish their respective tasks during the memory backtracking. (Sec. 2.2). Finally, we propose an iterative backtracking mechanism. Through a multi-stage iterative loop pipeline, it dynamically explores in-depth memories related to the question, thus efficiently collecting sufficient information both in breadth and depth. (Sec. 2.3). In brief, based on having a vague understanding of the entire long video, VideoLucy is designed to achieve accurate question answering by continuously expanding and delving deeper into its memory of the video, just like how humans recall past events.

### 2.1 Hierarchical Memory Structure

We demonstrate that a good memory structure capable of effectively representing abundant information of the entire long video should have the following characteristics.

1) Multi-level representation. Considering that the time ranges corresponding to the practical questions usually have a large span. For example, some questions focus on the single-frame understanding of an instantaneous moment, while others require a holistic understanding of the visual content within a large time range. Then, an good memory structure should conduct modeling for this multi-grained temporal scope, in other words, it should be equipped with the capacity for multi-level representation.

2) Comprehensive Information Coverage. One of the prominent characteristics of long videos is the presence of a vast amount of visual information. However, existing methods based on sparse frame sampling inherently result in significant information loss. Considering the uncertainty of the information quantity requirements of questions in practical applications, a good memory structure should be able to achieve comprehensive coverage of the information of the entire video.

Considering the above demonstration, we propose a new hierarchical memory structure, which, while effectively representing the visual content within multi-level temporal scopes, achieves comprehensive modeling coverage of the overall video information.

Specifically, for a video $V$ with $N$ frames, we use $f_i^{y_i}$ to represent each frame, where $i$ is the index of the frame in the video, and $y_i$ is the label indicating which clip the frame belongs to. Then we divide the video into $K$ short clips, and we use $v_k$ to represent each short clip, where $k$ is the index of the short clip. The meaning of $f_i \in v_k$ if $y_i = k$ is that if the clip label $y_i$ of $f_i$ is $k$, then this frame belongs to the $k$-th clip. Then for a single clip $v_k$, the memory of it is formulated as:

$$m_k = VidCap(v_k, p_k), \qquad (1)$$

where $VidCap$ represents any video MLLMs [2, 69] that are capable of providing a holistic text description of a given video based on the instruction prompt $p_k$. We can observe that this formula essentially determines the temporal scope of the memory by explicitly constraining the number of frames in each clip. When $K = 1$, this memory degenerates into a holistic overview of the entire video. When $K = N$, these memories represent detailed descriptions of each frame of the video.

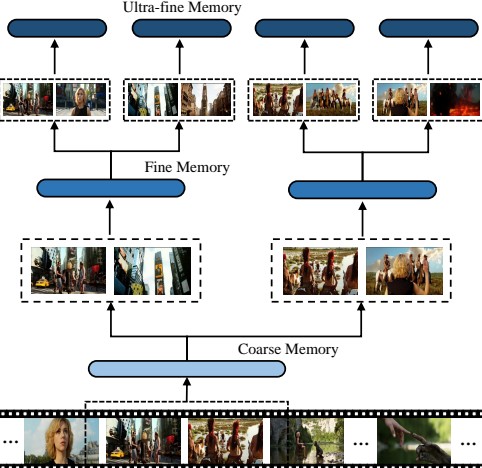

Figure 2: A toy example of hierarchical memory structure: For a video clip of a certain time segment, as the memory hierarchy deepens, the number of frames captured per second increases, and the time span covered by the memory shortens, thereby achieving multi-level video representations with comprehensive information coverage.

Then, we can obtain memories with different temporal perception scopes by specifying different values for $K$. We explicitly define three types of memories with successively decreasing temporal perception scopes to form a hierarchical memory structure. A toy example is shown in Fig. 2. From the shallow layer to the deep layer, they are respectively the long-range coarse memory, the short-range fine memory, and the frame-level ultra-fine memory. As clip division becomes denser, the number of text descriptions in these memories for the same-length video increases, achieving progressive memory detail granularity. With such a memory structure, we have not only achieved multi-level representation of long videos but also realized comprehensive information coverage.

## 2.2 Agents with Empowered Roles

As a common practice for an agent-based system [39, 66], an MLLM and LLM are empowered with respective roles to form different agents, through prompt engineering. These agents are required to accomplish their own tasks during the subsequent memory backtracking process. For the convenience, we list the roles of each agent as follows. More details can be found in the appendix.

**Captioning Agent.** Given a video clip and a caption instruction used to guide the focus, this agent with an MLLM [2] can provide a text description of the clip that meets the requirements. Its function is to serve as the "eye" of the entire system. By converting visual information into text content, it enables the system to perceive the video and extract memories.

**Localization Agent.** Given the current memory of the video (including text descriptions of various time periods) and the question, this agent, with the powerful text comprehension ability of the LLM [15] at its core, can provide a specified number of time periods that are most relevant to the question. This agent enables the system to filter out interfering memories and delve deeper into relevant memories, so as to achieve accurate and efficient question answering.

**Instruction Agent.** Given the current memory of the video, the relevant time period of interest from the localization agent, and the question, this agent can comprehensively understand the current memory, analyze the question-related key information that is missing in the given time period, and provide a guiding caption instruction. This instruction then enables the captioning agent to further delve into the memory, so as to supplement the question-related information.

**Answering Agent.** Given the current memory of the video and the question, this agent can, through in-depth reasoning and thinking, determine whether it can strictly and objectively provide a confident answer to the question based on the current memory. If it can answer, it will provide the answer. If it cannot answer, it will output an unconfident flag. This agent should not only be capable of answering questions but also tasked with deciding whether to further explore the memory.

## 2.3  Iterative Backtracking Mechanism

Although our proposed hierarchical memory structure achieves multi-level and comprehensive information coverage of the entire long video, it is obvious that if our system is simply built on the complete memories, it will undoubtedly introduce extremely high computational and storage costs, and is also likely to exceed the context limit of the LLM [15]. In addition, considering that users' questions usually focus on key time periods, a large amount of irrelevant deep memory will also turn into interfering information that cannot be ignored, which affects the system's performance.

Therefore, we propose a novel iterative backtracking mechanism. Through an agent-driven iterative loop, we continuously update the current memory initialized by sparse coarse memory, so as to dynamically explore the question-related memory in terms of both breadth and depth. This mechanism emulates the human recollection process and achieves a comprehensive search and integration of information relevant to the question with a relatively low resource cost.

---

**Algorithm 1** The Iterative Backtracking Mechanism

---

**Input:** The video $V$, question $Q$, captioning agent $CapAGT$, localization agent $LocAGT$, instruction agent $InsAGT$, answering agent $AnsAGT$ and the specified temporal scopes $T_c, T_f, T_{uf}$ corresponding to coarse, fine, and ultra-fine memory.
1: Implement sparse coarse memory initialization to obtain an initial current memory list $CM$.
2: Initialize a relevant set of time periods $S_{rt} = \{\}$.
3: Obtain the response based on the current memory $R = AnsAGT(CM, Q)$.
4: **while** $R$ is not confident **do**
5:     Locate the single most question-relevant time period $t = LocAGT(CM \setminus S_{rt}, Q)$ not in $S_{rt}$.
6:     Add this time period $t$ to the relevant set $S_{rt}$, i.e., $S_{rt} \leftarrow S_{rt} \cup \{t\}$.
7:     Analyze missing question-key info and provide instruction prompt $p = InsAGT(CM, Q, t)$.
8:     Obtain the video clip $V_t$ corresponding to this period $t$ from the video $V$.
9:     Divide $V_t$ into short clips $\{(t^i, V_t^i)\}_{i=1}^L$ by $T_d$, $|t| = T_c \Rightarrow T_d = T_f, |t| = T_f \Rightarrow T_d = T_{uf}$.
10:    Obtain the updated current-depth memory of this time period $m_c = CapAGT(V_t, p)$.
11:    Obtain the deeper memories of this time period $\{m_d^i\}_{i=1}^L = \{CapAGT(V_t^i, p)\}_{i=1}^L$.
12:    Update $CM$: $CM \leftarrow CM \cup \{(t, m_c)\} \cup \{(t^i, m_d^i) \mid i = 1, \cdots, L\}$.
13:    Obtain the response based on the updated current memory $R = AnsAGT(CM, Q)$.
14: **end while**
**Output:** The final response $R$ with a confident answer to the question $Q$.

---

***Sparse Coarse Memory Initialization.*** We maintain a dynamically updated current memory list $CM$, which is initialized with sparse coarse memory. Specifically, for a single video $V = \{f_i\}_{i=1}^N$, we first set a relatively large temporal scope $T_c$ corresponding to the coarse memory. Employing our captioning agent as the video captioner, we can obtain $K_c$ text descriptions $\{m_c^k\}_{k=1}^{K_c}$ according to Eq. 1, where $K_c = \lceil \frac{N}{T_c} \rceil$. The time period $t^k$ corresponding to each $m_c^k$ in the video are also recorded. Then, we initialize the current memory list and obtain $CM_{init} = \{(t^k, m_c^k)\}_{k=1}^{K_c}$. However, as stated previously, most of the memories are irrelevant to the questions. Therefore, we also adopt a sparse strategy. Specifically, for a question, we utilize the localization agent to find several most relevant time periods in $CM_{init}$, forming a time period set $S_t$. Subsequently, the updated current memory list can be obtained through filtering according to the time index by $CM_{sinit} = CM_{init} \cap S_t$.

***Question-guided Memory Exploration in Depth and Breadth.*** Generally speaking, just as it is difficult for humans to determine the details of past events based on vague memories, relying solely on the coarse-grained initial current memory list is not sufficient to provide a confident answer to the question. Therefore, our system is designed to conduct memory exploration in terms of depth (the detail level of the same time period) and breadth (different time periods) to further search for comprehensive question-relevant information. Specifically, given a question and the current memory list, we first let the localization agent locate the single most question-relevant time period. Then, we let the instruction agent analyze what question-related key information is lacking in the current text description of this time period, so as to provide a caption instruction. Finally, we instruct the caption agent to generate two types of text descriptions based on the video clip of this time period and the caption instruction. On one hand, it regenerates new text descriptions for the entire video clip (updating the current-depth memory). On the other hand, it divides the video clip into more short segments and generates text descriptions for each of them (exploring deeper memory). These descriptions will be used to update the current memory list. By iteratively executing this procedure,

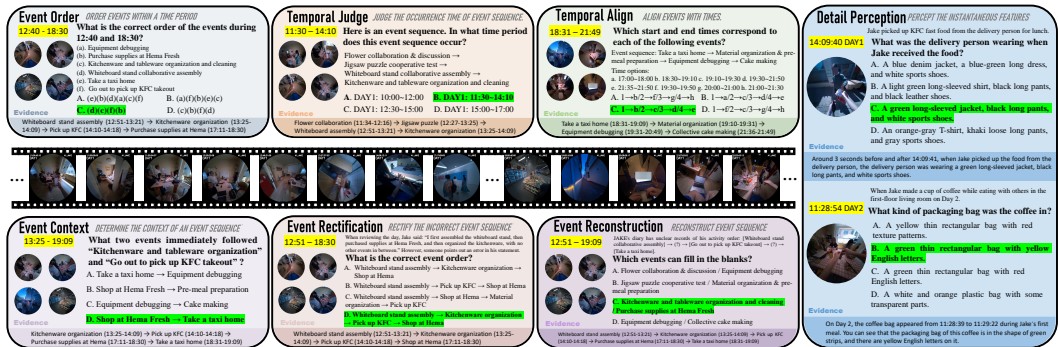

Figure 3: Question types and examples in the EgoMem benchmark. We design six question types to comprehensively assess the model's understanding of cross-temporal events in long videos. Additionally, a detail-perception task evaluates its ability to capture instantaneous visual features. All the questions are annotated manually, accompanied by sufficient evidence descriptions.

we obtain memories of different time periods augmented with question-relevant text descriptions. In other words, memory exploration in terms of both depth and breadth in achieved.

*Agent-driven Iterative Loop.* To efficiently answer questions regarding long videos and continuously collect question-relevant deep memories, we propose an iterative loop driven by multiple agents. The aforementioned two procedures, serving as the core functions of our system, are utilized in the agent-driven iterative loop. The specific algorithm process is shown in Alg. 1. In simple terms, we first obtain a memory list that is strongly related to the question through the proposed sparse coarse memory initialization method. Then, based on this memory list, we carry out multiple iterative memory explorations guided by the question, so as to fully search for and integrate the question-related detailed information across different time periods in the video. The iterative loop will stop only when the answering agent believes that the contained information in the updated memory list allows it to confidently answer the question. Then, it will output the corresponding confident answer. In practice, we set the maximum number of iterations to prevent getting trapped in an information search chain that causes timeouts. Through the cooperation of multiple agents, this iterative loop can efficiently achieve memory backtracking related to the question, so as to provide confident answers by searching and integrating objective clues as much as possible.

## 3 EgoMem Benchmark

We construct a new benchmark for ultra-long video understanding, namely EgoMem, aiming to measure the ability to model instantaneous (detail perception) and continuous (event understanding) memory of long videos. Based on the video resources of EgoLife [51], we manually annotate question-answer pairs that particularly focus on the understanding of cross-temporal events and the perception of instantaneous visual features for each day's long video. As shown in Fig. 3, for the event understanding, we design six different question types to conduct a comprehensive and effective evaluation of the model's performance in a real sense, and to avoid shortcuts. In addition, we manually annotate questions about the subtle visual features in those instantaneous time segments to assess whether the model can effectively cover detailed information. It contains 42 videos with an average duration of 6.33 hours and 504 questions. More details can be found in the appendix.

## 4 Experiments

**Implementation Details.** Our VideoLucy, as an agent-based system, only requires one LLM and MLLM for text comprehension and vision captioning. Unlike most other methods that call the APIs of expensive closed-sourced proprietary models [1, 42], we consistently use the open-sourced models Qwen-2.5-VL-7B [2] and DeepSeek-R1 [15] to ensure the reproducibility of the results and the low cost, if not specially stated. The temporal scopes $T_c$, $T_f$, $T_{uf}$ have distinct settings for different video benchmarks. We provide more details, especially the agent prompts, in the appendix.

**Evaluation Benchmarks and Metrics.** Following common practices, we conduct experiments mainly on three existing long video benchmarks. **MLVU** [67] is a comprehensive benchmark designed to evaluate both global and local video understanding, encompassing a diverse set of nine tasks. It includes videos ranging from 3 minutes to 2 hours in duration. **Video-MME** [11] contains

| Method | ER | EU | KIR | TG | Rea | Sum | Overall |
|---|---|---|---|---|---|---|---|
| *Proprietary Models* | | | | | | | |
| Gemini 1.5 pro [42] | 32.1 | 30.9 | 39.3 | 31.8 | 27.0 | 32.8 | 33.1 |
| GLM-4V-plus-0111 [13] | 46.2 | 47.8 | 54.1 | 42.7 | 46.5 | 37.9 | 48.7 |
| GPT-4o-20241120 [1] | 48.9 | 49.5 | 48.1 | 40.9 | 50.3 | 50.0 | 48.9 |
| *Leading Open-sourced MLLMs* | | | | | | | |
| TimeMarker-8B [5] | 42.8 | 39.1 | 34.9 | 38.7 | 38.2 | 48.8 | 41.3 |
| VideoLLaMA3-7B [58] | 45.8 | 42.4 | 47.8 | 35.9 | 45.8 | 36.2 | 45.3 |
| InternVL2.5-78B [7] | 43.8 | 42.0 | 42.1 | 36.8 | 51.0 | 37.9 | 43.6 |
| Qwen2-VL-72B [43] | 38.0 | 41.1 | 38.3 | 41.4 | 46.5 | 46.6 | 41.3 |
| ReTake-7B [45] | 49.8 | 46.2 | 52.9 | 45.0 | 45.8 | 27.6 | 47.8 |
| VideoChat-Flash-7B [24] | 51.1 | 46.0 | 49.0 | 38.9 | 48.5 | 34.5 | 48.2 |
| AdaReTaKe-72B [46] | 53.0 | 50.7 | 62.2 | 45.5 | 54.7 | 37.9 | 53.3 |
| *Agent-based Systems* | | | | | | | |
| VideoAgent [47] | 28.0 | 30.3 | 28.0 | 29.3 | 28.0 | 36.4 | 29.3 |
| VideoTree [48] | 30.3 | 25.1 | 26.5 | 27.7 | 31.9 | 25.5 | 28.8 |
| MemVid [56] | 53.4 | 40.6 | 46.3 | 34.9 | 43.2 | 28.1 | 44.4 |
| VCA [52] | 43.7 | 40.7 | 37.8 | 38.0 | 46.2 | 27.3 | 41.3 |
| **VideoLucy** | **54.3** | **59.8** | **75.6** | **51.7** | **55.9** | **49.1** | **58.8** |

| Method | M-Avg |
|---|---|
| *Proprietary Models* | |
| GPT-4V [1] | 49.2 |
| GPT-4o [1] | 64.6 |
| *Open-Sourced MLLMs* | |
| Video-CCAM-14B [10] | 63.1 |
| Video-XL-7B [37] | 64.9 |
| LLaVA-OV-72B [18] | 66.4 |
| LinVT-7B [12] | 68.9 |
| Aria-25B [19] | 70.6 |
| Oryx-1.5-32B [30] | 72.3 |
| VideoLLaMA3-7B [58] | 73.0 |
| VideoChat-Flash-7B [24] | 74.7 |
| *Agent-based Systems* | |
| VideoTree [48] | 60.4 |
| Video-RAG-7B [31] | 72.4 |
| VideoMind-7B [29] | 64.4 |
| **VideoLucy** | **76.1** |

Table 2: Comparison results on **LVBench**. Most are from the latest leading results of the official leaderboard (updated 20250515), and the results of the agent systems are from the corresponding papers.

Table 3: **MLVU** comparison. The latest leading results of the official leaderboard (updated 20250515) or respective papers.

2,700 manually annotated questions of 900 distinct videos with different durations: short (< 2min), medium (4min~15min), and long (30min~60min). We report its results under the without subtitle setting. **LVBench** [44] is designed for ultra-long video understanding, containing 1,492 questions of 99 videos, with an average duration of 4,101 seconds. It features a diverse set of 6 tasks, all supported by high-quality human annotations. Additionally, our proposed **EgoMem** benchmark is utilized for further performance comparisons. As the default setting, we use accuracy as the evaluation metric.

## 4.1 Main Comparison with Other Methods

**Comparison on Video-MME.** We compare our VideoLucy with latest leading methods on Video-MME [11] in Tab. 1. We classify the previous methods into three categories, the proprietary models, the open-source MLLMs trained with numerous image and video data, and the agent-based systems with eye-catching zero-shot capabilities. As it shows, our VideoLucy achieves a far-surpassing performance compared with other agent systems, with average accuracy 8.5% higher than that of the previous best, MemVid. In addition, our VideoLucy stands out in terms of long video understanding. Even though VideoLucy uses a 7B MLLM, it still achieves the best performance among all open-sourced MLLMs in this aspect, outperforming the best model Adaretake-72B by 1.8%, and is on a par with the leading commercial proprietary model Gemini 1.5 pro.

**Comparison on LVBench.** The outstanding ability of our VideoLucy has been further significantly demonstrated on the ultra-long benchmark LVBench [44]. As shown in Tab. 2, our Vide-

Table 1: **Video-MME** (w/o subs) comparison. The latest leading results of the official leaderboard (updated 20250515) or respective papers.

| Method | Short | Medium | Long | Avg |
|---|---|---|---|---|
| *Proprietary Models* | | | | |
| Gemini 1.5 Flash [42] | 78.8 | 68.8 | 61.1 | 70.3 |
| GPT-4o-20240513 [1] | 80.0 | 70.3 | 65.3 | 71.9 |
| Gemini 1.5 pro [42] | 81.7 | 74.3 | 67.4 | 75.0 |
| *Leading Open-sourced MLLMs* | | | | |
| Qwen2-VL-72B [43] | 80.1 | 71.3 | 62.2 | 71.2 |
| AdaReTake-72B [46] | 80.6 | 74.9 | 65.0 | 73.5 |
| InternVL2.5-72B [7] | 82.8 | 70.9 | 62.6 | 72.1 |
| LLaVA-OneVision-72B [18] | 76.7 | 62.2 | 60.0 | 66.3 |
| LLaVA-Video-72B [63] | 81.4 | 68.9 | 61.5 | 70.6 |
| VideoChat-Flash-7B [24] | 78.0 | 67.8 | 55.6 | 65.3 |
| VideoLLaMA3-7B [58] | 80.1 | 63.7 | 54.9 | 66.2 |
| LiveCC-7B [4] | 74.8 | 63.9 | 53.7 | 64.1 |
| ViLAMP-7B [8] | 78.9 | 65.8 | 57.8 | 67.5 |
| LinVT-7B [12] | 79.0 | 71.6 | 63.2 | 70.3 |
| *Agent-based Systems* | | | | |
| VideoAgent [9] | - | - | 46.4 | - |
| VideoTree [48] | 67.8 | 59.9 | 54.2 | 60.6 |
| DrVideo [32] | - | - | 51.7 | - |
| MemVid [56] | 73.9 | 63.1 | 55.0 | 64.0 |
| Video-RAG-7B [31] | 66.4 | 60.2 | 59.8 | 62.1 |
| VCA [52] | - | - | 56.3 | - |
| **VideoLucy** | **78.6** | **72.1** | **66.8** | **72.5** |

oLucy achieves performance far exceeding the previous best methods across all metrics. Its overall performance is 5.5% higher than the latest best method [46] on the official leaderboard, achieving 58.8% accuracy. In addition, it performs particularly outstandingly in key information retrieval (KIR), achieving an accuracy rate of 75.6%, significantly surpassing all previous models.

**Comparison on MLVU.** We also evaluate the performance of our VideoLucy on the comprehensive benchmark MLVU [67], which covers spanning video durations. As shown in Tab. 3, compared with the latest leading methods on the official leaderboard, our VideoLucy still achieves better overall

results. This indicates that our VideoLucy has excellent adaptability to videos of various durations and can effectively meet the situation where the video duration is unknown in practical applications.

**Comparison on EgoMem.** We also carried out a performance evaluation comparison on our proposed EgoMem benchmark. As shown in Tab. 4, we tested a series of the latest leading open-sourced video MLLMs. It can be seen that existing MLLMs perform poorly, and they are seriously lacking in cross-temporal understanding and detail perception abilities in the case of extremely long videos. Their evaluation results [43, 2, 7, 69] are only slightly better than the results of random guessing. However, our VideoLucy, can better understand cross-temporal and detailed information of extremely long videos through the memory backtracking mechanism. Compared with other methods, it achieves the best performance on EgoMem, with an overall accuracy rate of 56.7%, which is 10.3% higher than that of the latest ultra-long video understanding model, VideoChat-Flash [24].

Table 4: Evaluation results on our proposed **EgoMem** benchmark.

| Method | Event | Detail | Avg |
|---|---|---|---|
| *Leading Open-sourced MLLMs* | | | |
| Qwen2-VL-7B [43] | 38.9 | 25.8 | 32.3 |
| Qwen2-VL-72B [43] | 38.1 | 27.4 | 32.7 |
| Qwen2.5-VL-7B [2] | 36.5 | 27.8 | 32.2 |
| Qwen2.5-VL-72B [2] | 37.7 | 30.2 | 33.9 |
| InternVL2.5-8B [7] | 33.7 | 34.5 | 34.1 |
| InternVL2.5-78B [7] | 32.7 | 38.3 | 35.5 |
| InternVL3-8B [69] | 28.2 | 38.9 | 33.6 |
| InternVL3-78B [69] | 27.4 | 35.2 | 31.3 |
| VideoChat-Flash-7B [24] | 44.8 | 48.0 | 46.4 |
| *Agent-based Systems* | | | |
| VideoAgent [47] | 28.6 | 34.5 | 31.5 |
| VideoTree [48] | 30.2 | 36.1 | 33.1 |
| **VideoLucy** | **58.7** | **54.8** | **56.7** |

## 4.2 Ablation and Analysis

**Needle-in-A-Video-Haystack.** To further explore the model's ability to perceive the detailed instantaneous events in long videos, following the common practices [24, 62, 65], we conduct an evaluation experiment named "Needle-in-A-Video-Haystack". Specifically, we randomly select 10 long videos with durations ranging from 400s to 4000s from the existing benchmarks [67, 44]. Then, we insert 10s short video clips (needles) from the Internet at five arbitrary timestamps throughout each long video, from the beginning to the end. Then we feed the entire long video into the model and conduct a question-answering test regarding the content of these short clips. There are 4 questions for each clip, forming 20 questions for one long video. As shown in Fig. 4, the performance of our VideoLucy is significantly better than that of the existing leading models, and its results are almost unaffected by the video length. This indicates that our VideoLucy has a very strong ability to search for question-relevant details in long video understanding. More details are shown in the appendix.

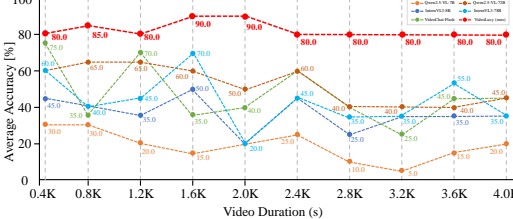

Figure 4: Results on the "Needle-in-A-Video-Haystack" evaluation with 10 long videos.

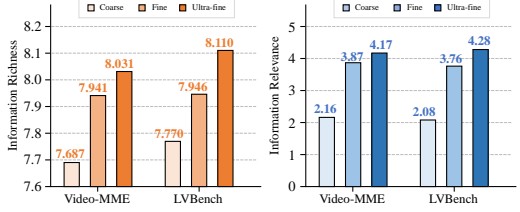

Figure 5: Evaluation on information richness and relevance in memory backtracking of VideoLucy.

**Information Richness and Relevance in Backtracking.** We experiment to explore the information richness of memories at different levels during memory backtracking and their relevance to the given questions. Specifically, we first select the test samples in Video-MME [11] and LVBench [44] that enter the deepest memory of VideoLucy. Then, we evaluate the information richness by calculating the average value of the Shannon entropy of the text descriptions of memories at all levels for each test sample. In addition,

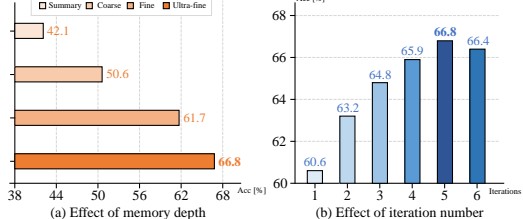

Figure 6: Effects of different memory depth and maximum iteration number. We evaluate the performance on Video-MME long split (w/o subs).

through prompt engineering, we make an LLM [35] serve as a relevance evaluator, enabling it to assess the relevance between the given text description and the question. As shown in Fig. 5, during the backtracking process, both the richness and relevance increase continuously, verifying the effectiveness of our method. More details are shown in the appendix.

**Effects of Different Components.** We conduct an experiment on the Video-MME [11] long split to explore the effectiveness of memories at various levels on the performance of the model. The

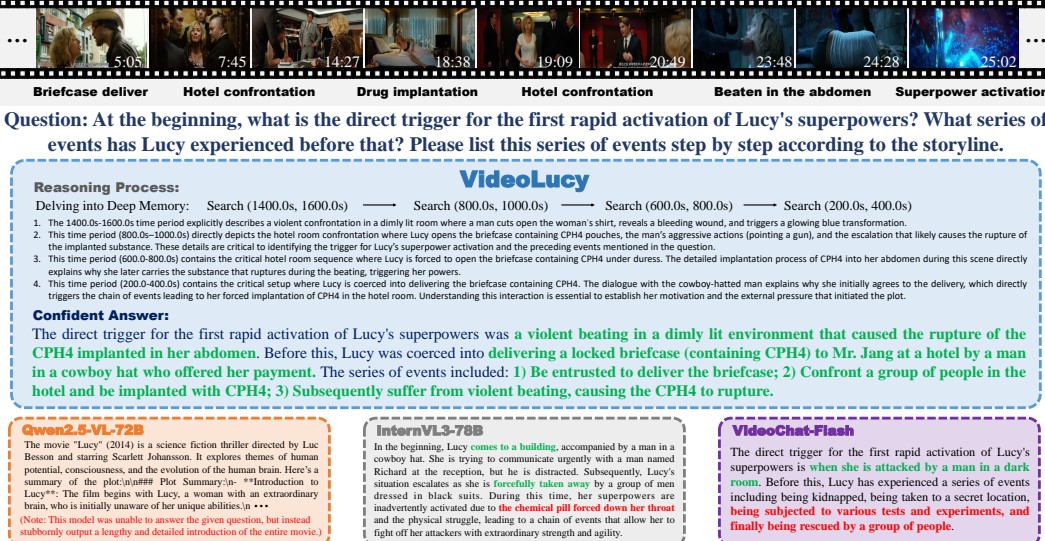

Figure 7: Qualitative comparison of event understanding in long videos. Compared with existing leading video MLLMs [2, 69, 24], our VideoLucy stands out in capturing and integrating cross-temporal events in long videos, along with explainable and comprehensive reasoning process.

experiment is divided into four groups: a model answering with only a rough video summary; a model relying on coarse-memory text; a model with fine-grained memory access; and a model with ultra-fine-grained memory access. As shown in Fig. 6 (a), delving into memories at all levels can effectively improve the model performance, and the optimal result is achieved when delving into the frame-level ultra-fine memory. In addition, we explore the impact of the maximum number of iterations in the loop. As shown in Fig. 6 (b), the best performance is achieved when the number of iterations is set to 5, which is set as our default value.

## 4.3 Qualitative Comparison

We conduct a qualitative comparison experiment using a long video of the movie Lucy as the test material. The experiment consists of two tasks. In the first task (Fig. 7), models [2, 69, 24] are required to address questions on cross-temporal event understanding. As it shows, our VideoLucy accurately captures inter-temporal event relationships, outperforming the other three models that often provide inaccurate or incomplete answers. In the second task (Fig. 8), models are asked to answer questions about the detailed information of momentary segments with fleeting events. As we can see, our VideoLucy performs well in extracting and presenting fine details, while the other two models struggle to capture nuanced information, yielding poor results. Additionally, for each question in both tasks, our VideoLucy not only provides the correct answer but also shows the

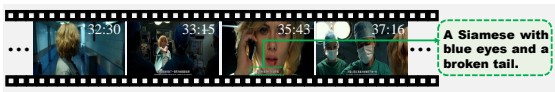

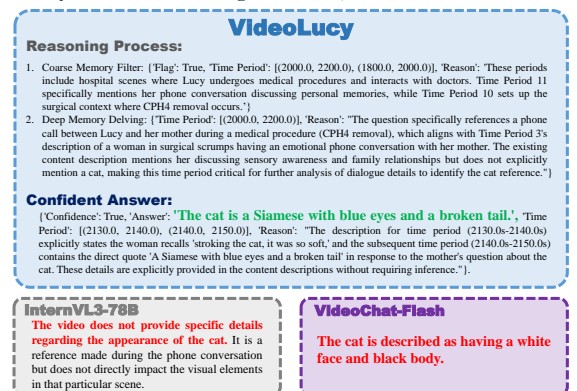

Figure 8: Qualitative comparison of detail perception in long videos. VideoLucy enjoys great superiority.

step-by-step process of how it arrives at that conclusion. This transparency helps users understand the model's decision-making logic in practice, enhancing result interpretability and trust.

## 5 Related Work

**MLLMs for Long Video Understanding.** Existing video MLLMs [22, 21, 26, 60, 36, 23] are generally built upon conventional image-based VLM frameworks [27, 68, 20]. They employ a visual

encoder to convert frames into tokens, which are then projected into an LLM's semantic space to support tasks such as video captioning, QA, and reasoning. For long videos, two main trends distinguish video MLLMs from standard VLMs.

First, frame sampling has gained attention as a critical component. Since processing all frames in long videos is computationally prohibitive, the focus is on selecting key frames relevant to the query or task. Several studies [41, 16, 55, 54, 53, 33, 40, 61] adopt attention-based mechanisms to model spatio-temporal relationships among frames. For example, [41] proposes Adaptive Keyframe Sampling (AKS), which optimizes both relevance to the prompt and video coverage under a fixed token budget, thereby improving QA performance and underscoring the value of information pre-filtering.

Second, token compression has emerged to address the large number of visual tokens produced by long videos, which can exceed LLM capacity and increase computational overhead. Researchers [25, 10, 28, 38, 49, 6, 57] have developed methods to compress tokens while preserving essential information. For instance, VideoChat-Flash [24] introduces a hierarchical token compression approach combined with a multistage learning strategy, achieving strong performance under high compression rates.

In addition, some recent works have introduced memory-augmented methods. LangRepo [17] pioneers a long-video understanding framework that maintains a structured language repository for LLMs. It iteratively updates multi-scale video chunks and prunes redundant text via write/read operations. Similarly, TTM [34] propose an efficient memory-augmented Transformer inspired by Neural Turing Machines. It uses an external memory to compress historical frames into compact tokens, limiting computational complexity as each new frame interacts only with memory tokens.

Nevertheless, these conventional video MLLMs still struggle to effectively represent ultra-long videos. Their reliance on sparse frame sampling often results in significant information loss and suboptimal responses in real-world scenarios.

**Agent-based Systems for Long Video Understanding.** Recent studies [50, 14, 3, 32, 9, 47, 66, 48] have introduced agent-based systems as a promising solution for long video understanding. Unlike traditional video MLLMs, which struggle with lengthy visual inputs, these systems utilize LLMs for reasoning, planning, and memory management. By employing LLMs to orchestrate MLLMs, they decompose long videos into short-video subtasks, iteratively retrieving and integrating question-relevant information. This approach mitigates length constraints and enhances comprehension.

For instance, VideoAgent [47] employs an LLM as a central agent that iteratively locates and consolidates key information using vision-language models. DrVideo [32] transforms long videos into text documents and applies agent-based loops to retrieve key frames and augment data. VideoTree [48] builds a query-adaptive hierarchical video representation via iterative keyframe refinement and tree-based aggregation, enabling efficient LLM reasoning.

However, as noted, these systems typically reason over individual frames, overlooking temporal context, and often adopt sparse sampling to reduce captioning costs—potentially missing critical information. These limitations highlight the need for improvement, motivating our VideoLucy.

# 6 Conclusion

We introduced VideoLucy, a novel deep memory backtracking framework. Inspired by human recollection processes, VideoLucy employs a hierarchical memory structure with progressive granularity and an agent-based iterative backtracking mechanism. This design enables the system to dynamically and comprehensively mine question-relevant information from the entire video, starting from a coarse overview and progressively delving into finer spatiotemporal details, thereby achieving robust temporal understanding and unprecedented fine-grained perception. Furthermore, we presented EgoMem, a new benchmark for evaluating temporal and fine-grained understanding in extremely long videos. Extensive experiments demonstrate that VideoLucy achieves state-of-the-art performance across multiple benchmarks, significantly outperforming previous methods and even surpassing powerful proprietary models. Our work validates the effectiveness of a structured, human-like memory mechanism for complex video understanding and paves the way for future research in this field.

# 7 Acknowledgments.

This work was supported by the National Natural Science Foundation of China No.U22B2053, and the Hubei Provincial Natural Science Foundation of China No.2022CFA055.

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

# A  Appendix

In this appendix, we elaborate on the following:

**Technical Details.** Specifically, we demonstrate the distinct settings of our VideoLucy for different video benchmarks with varying video length distributions. These settings include the configuration of memory temporal ranges at all levels, frame sampling rate settings, etc. On the other hand, we describe the carefully designed prompt details for different agents.

**EgoMem Benchmark Details.** We provide a comprehensive elaboration on the meticulous and step-by-step process of manual annotation, as well as the detailed annotation content of question-answer pairs across each representative category.

**Ablation Study Details.** We elaborate on the specific experimental details of the two ablation studies in the main text. On one hand, for the Needle-in-A-Video-Haystack experiment, we explain the sources of the 10 selected long videos and present the "needles" we constructed, which consist of five short videos and 20 question-answer pairs. On the other hand, for the Information Richness and Relevance in Backtracking experiment, we display the code for information entropy calculation and the prompts designed for the relevance evaluator.

**Additional Experiments.** We evaluate the performance of our VideoLucy when using different MLLMs and LLMs to validate the excellent adaptability of our framework.

**Failure Case Analyses.** We present and analyze additional specific failure cases of VideoLucy's performance in long video understanding.

**Broader Impacts.** We discuss the broader societal impacts of long video understanding technology.

**Limitation.** We discuss the limitation of our VideoLucy framework.

**Inference Time.** We discuss the inference time of our VideoLucy framework.

## A.1  Technical Details

### Distinct Settings for Different Benchmarks

During the sparse coarse memory initialization of VideoLucy, for all benchmarks, we uniformly instruct the localization agent to first identify the three most relevant time periods. Additionally, considering that scenes with event continuity may be split into multiple periods, we adopt a neighborhood expansion strategy: the two time periods adjacent to a period provided by the agent are also included in the relevant time periods. As a result, the number of time periods in the sparse coarse memory ranges from three to nine. Additionally, the distinct settings are described as follows:

**MLVU**. Since MLVU is a comprehensive video understanding benchmark with a wide range of video durations, we divided it by video length: 0-600s (short), 600-1200s (medium), 1200-3600s (long), and >3600s (extra-long). Temporal scopes for memory: Short: $T_c = 30s$, $T_f = 5s$, Medium: $T_c = 60s$, $T_f = 5s$, Long: $T_c = 100s$, $T_f = 10s$, Extra-long: $T_c = 200s$, $T_f = 10s$. All use $T_{uf} = 1s$. The frame sampling rate for coarse-grained memory is 1 FPS, while the others are 2 FPS.

**Video-MME**. This benchmark has been divided into short, medium, and long splits by the original authors. Following convention, we adopt the official default splits. Temporal scopes for memory: Short: $T_c = 5s$, $T_f = 1s$, Medium: $T_c = 50s$, $T_f = 5s$, Long: $T_c = 100s$, $T_f = 10s$. All use $T_{uf} = 1s$. For medium and long videos, the frame sampling rate of coarse-grained memory is 1 FPS, while the other two are 2 FPS. For short videos, the frame sampling rate of coarse-grained memory is 2 FPS, and the other two are 4 FPS.

**LVBench**. It is a benchmark specifically designed for long video understanding, yet it still encompasses a relatively wide range of video durations. We also divide it by video length: 1800-3600s (short), 3600-5400s (medium), and >5400s (long). Temporal scopes for memory: Short: $T_c = 100s$, $T_f = 10s$, Medium: $T_c = 150s$, $T_f = 10s$, Long: $T_c = 200s$, $T_f = 10s$. All use $T_{uf} = 1s$. The frame sampling rate for coarse-grained memory is 1 FPS, while the others are 2 FPS.

**EgoMem**. Since each video in this benchmark is extremely long (with an average duration of 6.33 hours) and their lengths are roughly uniform, we do not perform any division on this benchmark. For each video, the temporal scopes are set as follows: coarse-grained memory ($T_c = 800s$), fine-grained

The following provides a rough description of what's shown in the video during different time periods:
{*Current Memory List (Time Period + Description)*}
Now, a question has been raised regarding this video.
{*Question*}
Please read the given video content descriptions and the question in depth.
Since most of these descriptions are rather rough and some detailed information is lost, my task is to try my best to find the time periods related to the given question, and then provide more detailed descriptions of the video content of these time periods.
In order to assist me in completing my task, your task is to:
Based on the provided rough video descriptions, determine whether the given question allows me to provide a more confident answer by further observing the video content of three time periods.
If so, you should find out the time periods related to the question as much as possible and provide these relevant time periods so that I can review the content information of these video segments again to obtain more information and answer the question better.
For example, since there is no need for an overall understanding of large video segments, the following questions can obtain more accurate answers by re-observing the video segments of three time periods:
(i) What color is Putin's tie between the interview with Antony Blinkoen and interview with Marie Yovanovitch?
(ii) How does the goalkeeper prevent Liverpool's shot from scoring at 81:38 in the video?
(iii) Who smashes the magic mirror?
On the contrary, for example, because an overall understanding of large video segments is required, it is difficult to obtain more accurate answers to the following questions by merely observing two video segments:
(i) What happens in the second half of the game?
(ii) What is the video about?
(iii) Which places has the protagonist of this video been to in total?
You should output in a strictly standardized dictionary format containing three key-value pairs:
"Flag": A bool. If you are very confident that you can provide the time periods according to the above requirements, set it as True. Otherwise, set it as False.
"Time Period": A list. If "Flag" is True, fill in the list with the most relevant three time periods, in the tuple format (start time, end time). If "Flag" is False, fill in "No Time Periods."
"Reason": A String. Show me your reasons for the time periods you provided.

Table 5: **The designed prompts for the localization agent** in the sparse coarse memory initialization stage. Given the current memory list and the question, we obtain the most relevant three time periods.

memory ($T_f = 80s$), and ultra-fine-grained memory ($T_{uf} = 8s$). The frame sampling rates are respectively set to 0.25 FPS, 0.5 FPS, and 1 FPS for the three memory types.

**Agent Prompts.**

**Captioning Agent**. The primary function of this agent is to serve as the system's "eye," capable of describing a given video clip according to the requirements of specified instructions. For all coarse memory extraction, which is unrelated to the question, the instruction prompt is uniformly set as: *"Please observe and understand the given video carefully. Describe all the details of this video as comprehensively as possible in a smooth and coherent passage. Do not omit any details or prominent information. In addition, if there are any texts, subtitles, text overlays, or voice-overs in the video, you must explicitly and in detail describe them."* For memory extraction at other levels, the instruction prompts are dynamically generated by the instruction agent based on the question.

**Localization Agent**. The main function of this agent is to locate the temporal segments in the current memory (including text descriptions of various time periods) that are relevant to the question. This agent is frequently used in sparse coarse memory initialization to find the three most question-relevant temporal segments. Considering that some questions require a comprehensive understanding of the entire video for answering, we have designed additional functionalities for this agent during the initialization process, as follows: if the given question can be answered by re-reviewing several temporal segments, these segments will be provided; if not, no sparse initialization will be performed. The prompt of this agent in the initialization phase is shown in Tab. 5. Additionally, this agent is also used to search for the single most question-relevant and interesting temporal segment in the current memory list during the memory backtracking process, after which the subsequent instruction agent will generate corresponding instructions for retrieving supplementary information. To minimize LLM calls, we merged these two processes into one, enabling the functions of the above two agents to be achieved simultaneously through a single designed prompt, as shown in Tab. 6.

There is currently a video with a total duration of *{video length}* seconds.
The following gives a general description of what is shown in the video during certain time periods:
{*Current Memory List (Time Period + Description)*}
Now, a question has been raised regarding this video.
{*Question*}
Please read the given video content descriptions and the question in depth.
You do not need to answer this question.
Your first task is to identify, based on the video content in each time period, the single time period that is most relevant to the question and that you think requires further elaboration of its video content details to make the answer to this question more explicit.
Notably, you need to select the most relevant one from the time periods other than the following time periods:
{*Already Searched Time Periods*}
In addition, assume there is now a caption model that can describe a given video according to your instruction. Your second task is to consider what detailed content in the video of the time period you have selected you want the model to focus on describing, and provide your instruction.
For example, assume that the entire video segment is about an offensive play in a certain football game, and you want to focus on the passing situation of the football during this offensive play. The instruction you give to the model could be:
Please observe all the details in this video very carefully and provide a detailed and objective description of what is shown in the video. If this video is about an offensive play in a football match, you should focus particularly on the passing situation of the football during this offensive play.
Note that you should organize your instruction by referring to the language expressions in the above example.
You should output in a strictly standardized dictionary format containing three key-value pairs:
"Time Period": A list. Fill with the single most relevant period, in the tuple format (start time, end time).
"Instruction": A String. This string must be enclosed in double quotes. Show me the instruction you want to give to the caption model for the second task.
"Reason": A String. This string must be enclosed in double quotes. Show me your reasons for the time period and instruction you provided.

Table 6: **The designed prompts for the localization and instruction agents** in the memory backtracking stage. Given the current memory list and the question, we obtain the single most relevant period and corresponding instruction.

**Instruction Agent**. Given the current memory list and a specified time period, this agent generates an instruction for the captioning agent to obtain question-relevant supplementary information. This agent can be merged with the localization agent, and their combined prompt is shown in Tab. 6.

**Answering Agent**. This agent can deeply reason and analyze based on the current video memory and the question to determine whether it can strictly and objectively provide a confident answer using the existing memory. If an answer can be provided, it will output the result; if not, it will generate an unconfident flag. In addition to answering questions, this agent is also responsible for deciding whether to further explore the memory. The prompt design for it is shown in Tab. 7.

## A.2 EgoMem Benchmark Details

**Manual Annotation Process**

***Build a sequence of events that occur throughout the day***. We require annotators to carefully review the daily videos in EgoLife, meticulously record the names of events that happen each day, and label each event with start and end times accurate to the minute, along with specific descriptions. Subsequently, arrange the entire day's events in chronological order to form a complete event sequence, ensuring that the timeline is continuous and logically consistent.

Then, the question-answer pairs for each task are annotated by referring to these event sequences.

For **event understanding task**, we adopt the following two steps to annotate question-answer pairs.

***1) Event subsequence extraction.*** A continuous target time period is selected from the complete event sequence of a day, and then 4 to 6 events are randomly extracted from all events corresponding to that period. It is worth noting that the continuity requirement for event subsequences needs to be adjusted according to question types: some question types (such as time period judgment and mask

| |
|---|
| There is currently a video with a total duration of *{video length}* seconds. |
| The following gives a general description of what is shown in the video during certain time periods: |
| {*Current Memory List (Time Period + Description)*} |
| Now, a question has been raised regarding the content descriptions of this video. |
| {*Question*} |
| Please read the given video content descriptions and the question in depth, and determine whether you can accurately answer the given question solely based on the currently provided descriptions. |
| If you can answer it with absolute confidence, please answer this question and provide the time periods of the video content you are referring to. The answer you provide must have completely and absolutely objective support in the video descriptions. Do not make inferences arbitrarily. |
| If you think the current content descriptions of the video are still insufficient to accurately answer the question, please do not answer it and give me your reason. |
| Please output in a strictly standardized dictionary format containing four key-value pairs: |
| "Confidence": A boolean value. Set it to True if you are certain about the answer, and False if not. |
| "Answer": A string. This string must be enclosed in double quotes. When "Confidence" is True, fill in the answer content; when "Confidence" is False, fill in "No Answer". |
| Time Period": A list. When "Confidence" is True, fill in the list with time periods corresponding to the answer, each in the format of a tuple (start time, end time); when "Confidence" is False, fill in "No Time". |
| "Reason": A String. This string must be enclosed in double quotes. Show me your reasoning about your judgment. You need to ensure and check that your reasoning must be able to absolutely support your answer. |

Table 7: **The designed prompts for the answering agent.**

prediction) require events to be continuous, while question types such as event sorting and time point correspondence allow events to be discontinuous.

*2) Designing six types of question-and-answer pair*. Event Order: introduce interfering events from other time periods, mix them with sub-sequence events, and then ask for chronological ordering within the time range specified in the question stem. Temporal Judge: omit the time information of event sub-sequences and require providing the actual occurring time period. Temporal Align: separate events from their time periods, mix in modified incorrect times or times from other events as distractors, and require matching each event with the correct time period. Event Context: based on 2-4 consecutive events, require inferring their preceding or following contexts. Event Rectification: modify the original sequence and require identifying and correcting erroneous events. Event Reconstruction: require completing the obscured events in a continuous sequence.

By having annotators carefully carry out the above steps for a full-day video, we can obtain six question-and-answer pairs focusing on different aspects of event understanding.

For **detail perception task**, the annotation process is simpler. We manually identify instantaneous visual features that appear in an extremely short timeframe from the already labeled event sequences, then create questions targeting these features. Highly confusing distractors are artificially designed, and each question stem is composed of background information plus a specific query. Each full-day video annotation includes six question-and-answer pairs for detail perception.

### Detailed Annotation Content

Taking the video from Day 1 of A6 SHURE in the benchmark as an example, we will demonstrate the detailed content of our manual annotation in this subsection.

First, we carefully reviewed the full-day video, then annotated the event names, start and end times, and detailed content of the events that occurred on this day, as shown in Tab. 8.

Then, based on these events, we manually annotated six question-and-answer pairs for event understanding (Tab. 9) and six question-and-answer pairs for detail perception (Tab. 10).

We can see that the QA pairs we have annotated are of high quality, effectively measuring the model's ability to understand cross-temporal relationships in long videos and capture details.

### A.3   Ablation Study Details

For the **Needle-in-A-Video-Haystack** experiment, the 10 long videos are from LVBench and MLVU respectively. The specific video sources and their lengths are described as follows. LVBench: EwskdNETNx8 (3600s), JlrzSvCsIjE (2791s), NzCO0G8AGLU (2415s), O14bbpvy2x0 (2005s),

**Equipment Assembly [11:09:45-11:23:49]:**
A6 SHURE collaborated with other members in the living room to complete the assembly of electronic equipment. The man in a blue short-sleeved shirt distributed equipment to other members and provided on-site guidance.
**Activity Planning Whiteboard Meeting [11:23:49-12:18:55]:**
A6 SHURE was responsible for writing content on the whiteboard while other members discussed beside him, finalizing invited guests, activity content, and timelines.
**Flower and Greeting Card Handling [11:34:37-12:00:17]:**
The man in a blue short-sleeved shirt brought flowers and greeting cards, and the woman in a white top, the woman in a blue long dress, and the woman in a white short-sleeved shirt handled them together.
**Jigsaw Puzzle Assembly [12:18:55-13:26:23]:**
A6 SHURE and other members gathered around a table to assemble a wooden circular jigsaw puzzle.
**Whiteboard Assembly [12:50:54-13:17:28]:**
A6 SHURE, the man in a blue short-sleeved shirt, and the woman in a white short-sleeved shirt assembled the whiteboard stand together and hung up the whiteboard.
**Second-Floor Chat [13:17:28-13:26:23]:**
A6 SHURE went to the second floor to chat with the man in a blue short-sleeved shirt and the woman in a white short-sleeved shirt. During this period, A6 SHURE made two cups of pour-over coffee.
**Lunch Takeout Ordering [13:26:23-14:18:34]:**
A6 SHURE and the man in a blue short-sleeved shirt jointly operated a mobile phone to order takeout, while others discussed and wrote lunch choices on the whiteboard.
**Supermarket Shopping [17:18:39-18:37:08]:**
A6 SHURE and other members purchased beverages, hot pot bases, and other ingredients in the supermarket.
**Metro Ride [18:37:08-18:47:08]:**
A6 SHURE navigated and walked to the metro station, then took the metro.
**Classroom Self-Study [20:02:32-20:52:44]:**
A6 SHURE walked to his girlfriend's classroom and studied with her there.
**Dinner at Off-Campus Noodle Shop [20:52:44-22:09:45]:**
A6 SHURE and his girlfriend departed from the classroom, walked to an off-campus noodle shop for dinner, and then walked back to the classroom.

Table 8: **The manually annotated event sequence** for A6 SHURE Day 1 in EgoMem.

pXD3txG2bVQ (3205s), YlQugR7KSKg (3996s). MLVU: 1 (401s), ego_81 (1250s), movide101_51 (796s), order_92 (1579s). The durations of these 10 videos approximately conform to a uniform distribution from 400 seconds to 4000 seconds. In addition, we obtained 5 short videos each with a duration of 10 seconds from the Internet and manually annotated each of these 5 short videos with 4 question-answer pairs about detail perception to form the "needles", as shown in Fig. 9.

```
def calculate_entropy(text):
    words = text.lower().split()
    word_counts = Counter(words)
    total_words = len(words)
    entropy = 0
    for count in word_counts.values():
        probability = count / total_words
        entropy -= probability * math.log2(probability)
    return entropy
```

Table 11: **The code of Shannon entropy calculation**.

For the **Information Richness and Relevance in Backtracking** experiment, the calculation code for Shannon entropy is shown in Tab. 11, and the designed prompt for the relevance evaluator is in Tab. 12.

## A.4 Additional Experiments

To verify the good adaptability of our VideoLucy framework to various MLLMs and LLMs , we conducted an ablation experiment to explore the performance of the entire agent system when different MLLMs and LLMs are replaced. To reduce experimental costs, we randomly selected 200 question-answer pairs from LVBench to form the test set for this ablation experiment. The

experimental results are shown in Tab. 13. We can see that our VideoLucy demonstrates excellent adaptability to various MLLMs and LLMs. Combining different MLLMs and LLMs consistently achieves good performance, indicating that VideoLucy is a highly robust framework for long video understanding agent systems. In addition, more powerful MLLMs and LLMs generally yield more superior performance. For instance, Qwen2.5-VL-72B demonstrates a significant performance advantage over its 7B counterpart, while DeepSeek-R1 also outperforms Qwen2.5-72B to a relatively greater extent. This indicates that with the advancement of MLLM and LLM technologies, the performance of VideoLucy can be consistently enhanced.

| MLLMs | LLMs | Acc. |
|---|---|---|
| Qwen2.5-VL-7B | DeepSeek-R1 | 57.5 |
| Qwen2.5-VL-72B | DeepSeek-R1 | 64.0 |
| InternVL2.5-8B | DeepSeek-R1 | 54.5 |
| InternVL2.5-78B | DeepSeek-R1 | 65.5 |
| Qwen2.5-VL-7B | Qwen2.5-72B | 48.0 |
| Qwen2.5-VL-72B | Qwen2.5-72B | 52.0 |
| InternVL2.5-8B | Qwen2.5-72B | 44.5 |
| InternVL2.5-78B | Qwen2.5-72B | 54.0 |

Table 13: **Ablation study** on the performance with different MLLMs and LLMs.

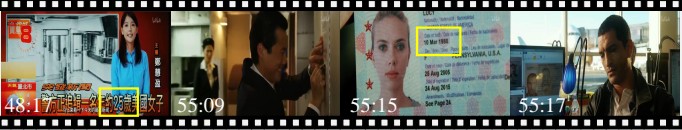

**How old is Lucy? What is her specific date of birth?**

**VideoLucy**

**Answer:**
Lucy is about **25 years old**, but her specific date of birth cannot be confidently inferred from the existing video descriptions.

**Reason:**
In the time period (2890.0s-2900.0s), a news broadcast clip appears, in which a Chinese subtitle translates to "The police are pursuing an American woman aged about 25." Therefore, it can be inferred that Lucy is approximately 25 years old. However, none of the detailed descriptions explicitly state Lucy's specific date of birth, so a confident answer cannot be given regarding it.

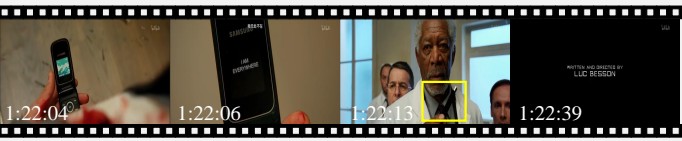

**At the end of the movie, what is the professor wearing a white coat holding in his hand?**

**VideoLucy**

**Answer:**
The professor is holding a black fountain pen in his hand.

**Reason:**
In the final scene of the movie, a description of a time period (5290s-5300s) directly states that an elderly man in a white lab coat is holding a black fountain pen in his hand, so a confident answer can be given.

Figure 10: **Failure Case Analyses.** In Example 1, since no clues related to the specific date of birth are reflected in the coarse-grained memory, and the question has no scene prompts in this regard, VideoLucy has difficulty locating the relevant time period within five iterations. In Example 2, due to the hallucination phenomenon occurring in the MLLM, which acts as the system's "eye", the object was not correctly understood and identified, leading VideoLucy to output an incorrect answer as well.

### A.5 Failure Case Analyses

Like existing agent systems, the performance of our VideoLucy is also constrained to a certain extent by the inherent capabilities of the MLLMs and LLMs within the system. We demonstrate

two representative failure cases in Fig. 10. In the first case, during the initial extraction of the video's coarse-grained memory, detailed information—such as the passport details in the frame—was overlooked, with only a general description ("a passport appears") retained. When users queried Lucy's exact date of birth without contextual cues, the LLM's inherent limitations prevented it from inferring that the date might be in the passport. Consequently, the system failed to locate the passport scene within five retrieval cycles, lacking any relevant information to answer the query accurately. In the second case, the user's explicit scene prompt ("the end of the movie") enabled the system to directly locate the target time period, followed by fine-grained information retrieval. However, due to MLLM hallucination, the model misidentified the black flash drive in the professor's hand as a black fountain pen—attributed to their visual similarity. Then, the system's default trust in MLLM outputs led to an erroneous response. Evidently, the system's response failures fundamentally stem from the current limitations of MLLMs and LLMs. However, this can be significantly mitigated by integrating more capable models. Thus, advancing MLLM and LLM technologies is imperative for future improvements, which will directly enhance VideoLucy's capabilities in tandem.

## A.6 Broader Impacts

The development of long video understanding technology has brought profound changes to various fields of society. Its positive impacts are mainly reflected in education, healthcare, the content industry, scientific research and etc. In the field of education, this technology supports personalized learning. Students can ask real-time questions about teaching videos and receive answers, breaking through the limitations of time and space. In medical scenarios, doctors can quickly analyze the key information in surgical videos to assist in case review and teaching. In addition, users of video platforms can accurately retrieve content through natural language, significantly enhancing the consumption experience. Researchers can also use this technology to automatically analyze experimental videos, improving data processing efficiency.

However, the application of this technology has also given rise to a series of negative impacts. Privacy and data security risks are the most pressing concerns, as the analysis of videos in private settings may lead to the leakage of personal sensitive information. The authenticity of information also faces challenges, as the combination of deepfake videos and this technology can easily lead to the spread of rumors. At the same time, the structure of the labor market has been impacted, resulting in a decrease in the demand for jobs such as content moderation and media editing. Ethical and legal issues, including algorithmic bias and copyright disputes, also urgently need to be addressed.

To rationally utilize long-video understanding technology, a multi-level governance framework integrating technology, policy, and social coordination is needed. Technologically, develop privacy-preserving computing like federated learning and differential privacy to safeguard data during analysis. Implement a multi-modal cross-validation mechanism, verifying text, images, and audio, to boost content understanding accuracy and anti-forgery capabilities. Policy-wise, expedite legislation to define video data collection, storage, and usage boundaries. Establish an algorithm registration and transparency review system, mandating technology users to disclose core algorithm logic and data sources. Set up specialized regulators to rigorously assess public-interest applications, such as security monitoring and medical video analysis. Socially, reinforce technological ethics education, heighten public awareness of tech limitations, and encourage enterprises, academia, and social groups to co-create industry self-regulatory guidelines. These actions will foster technology development to benefit society while respecting privacy and ensuring fairness.

## A.7 Limitation

Although our VideoLucy, benefiting from the hierarchical memory structure and iterative backtracking mechanism, can effectively understand long videos better than existing methods. However, like almost all existing agent-based systems, its performance is still affected by the respective capabilities of each agent in the system.

The accuracy of video MLLMs, which act as the system's "eyes", in understanding short videos will directly affect VideoLucy's subsequent backtracking process. Our VideoLucy utilizes existing video MLLMs to describe given video clips and extract corresponding memories. Obviously, if an MLLM with limited capabilities fails to effectively understand a given video and provides incorrect descriptions (i.e., hallucinations occur), this will negatively impact the factual consistency of the

memories stored in VideoLucy, thereby becoming unavoidable interfering information in the system. However, improving the capabilities of the captioner can theoretically significantly enhance system performance. As a transformation-friendly framework, any MLLMs can be easily integrated into our VideoLucy. That is, with the development of MLLM technology, the performance of our VideoLucy will also achieve gradual improvement.

As the core of the entire system's reasoning, the LLM's capabilities in text understanding and instruction following will also affect the system's performance. Our VideoLucy leverages existing LLMs to understand memories (textual descriptions) and requires them to complete corresponding tasks, such as question answering, key time positioning, missing information analysis, etc. Similarly, if an LLM with limited capabilities is unable to fulfill the above responsibilities and performs erroneous analysis and processing of the text, it will affect the system's performance to a certain extent. Additionally, the maximum video length that VideoLucy can process is also affected by the LLM's maximum context limit, which serves as a bottleneck for super-long videos longer than existing benchmarks, such as hundreds of hours. However, like other agent systems, our VideoLucy is a training-free framework, allowing any LLM with improved capabilities to be seamlessly integrated into it. Therefore, with the development of LLM technology, both the performance of our VideoLucy and the maximum video length it can process will improve steadily.

Additionally, like existing video agent systems, since our VideoLucy requires multi-round analytical processing using LLMs, it generally takes relatively more time to derive confident answers compared to end-to-end video MLLMs. However, considering that the memories obtained in our framework can be stored, for multiple different questions about the same video, we can fully leverage the video's pre-existing memories to achieve faster responses, instead of repeatedly performing inference on the video's information as existing video MLLMs do. This can reduce the time overhead of reasoning to a certain extent, which is conducive to deployment in practical applications.

In future work, we will focus on exploring how to reduce the impact of unavoidable noise information in agent systems and how to improve the operational efficiency of agent systems through better framework design while ensuring performance. We believe that our VideoLucy, as an agent system with comprehensive dynamic memory of long videos, can bring new inspirations to this community.

### A.8 Inference Time

We acknowledge the importance of inference efficiency in practical applications and provide an analysis of VideoLucy's inference time compared to other agent-based methods.

The inference overhead of VideoLucy primarily stems from two components: the MLLM processing short video clips and the LLM reasoning over long textual contexts. Both components can be effectively accelerated using modern inference frameworks. In our implementation, we deployed Qwen2.5-VL-7B locally on 8 A100 GPUs using vLLM, achieving efficient batch processing that generates textual descriptions for one-hour videos within dozens of seconds. For the LLM component, we utilized the official API of DeepSeek-R1 due to the high cost of local deployment.

To ensure fair comparisons across different agent-based systems, we adopt the number of LLM calls as our primary efficiency metric. This approach effectively eliminates confounding factors from hardware configurations and deployment environments, providing an objective basis for comparing methodological efficiency.

We conducted a controlled experiment using 42 question-answer pairs randomly selected from the EgoMem dataset, with the maximum number of LLM calls limited to 20. The results demonstrate that VideoLucy requires significantly fewer LLM calls (6.3 on average) compared to VideoAgent (14.1) and VideoTree (9.6), indicating a substantial advantage in inference efficiency. This efficiency gain stems from our optimized reasoning framework that minimizes unnecessary iterative processes while maintaining high answer quality.

**[Event Order]**
A6 SHURE forgot the events he experienced between 11:00 noon and 1:30 pm on DAY1 (a. Device assembly; b. Activity planning whiteboard meeting; c. Jigsaw puzzle; d. Supermarket procurement). If you were to sort the events that occurred during this period, what would be the correct order?
A. (a)(b)(c)    B.(a)(c)(b)    C. (b)(a)(c)    D. (c)(a)(b)
Evidence: Device assembly (11:09:45-11:23:49) → Activity planning whiteboard meeting (11:23:49-12:18:55) → Jigsaw puzzle (12:18:55-13:26:23). Supermarket procurement occurred from 17:18:39 to 18:37:08, not within 11:00-13:30, so options with d are excluded. The order in option A fits the time logic.

**[Temporal Judge]**
When summarizing his work, A6 SHURE wrote the sequence: [Device assembly → Activity planning whiteboard meeting → Jigsaw puzzle]. What time range should A6 SHURE annotate?
A. DAY1: 11:00 12:00    B. DAY1: 11:00 13:00    C. DAY1: 11:00 13:30    D. DAY1: 11:00 14:00
Evidence: Device assembly (11:09:45-11:23:49) → Activity planning whiteboard meeting (11:23:49-12:18:55) → Jigsaw puzzle (12:18:55-13:26:23). Total span: 11:09:45-13:26:23, matching C (11:00-13:30).

**[Temporal Align]**
A6 SHURE only recorded what he did, but did not record the start and end times of each thing. Now there are some time periods for A6 SHURE to choose from. Event sequence: [Jigsaw puzzle → Whiteboard assembly → Second-floor chat]. Time options: a. 12:00-13:00 b. 12:15-13:30 c. 12:50-13:20 d. 13:15-13:30 e. 13:00-13:30 f. 12:18-13:26 Can you help A6 SHURE find the correct time period for each event?
A. 1→f / 2→c / 3→d    B. 1→a / 2→c / 3→e    C. 1→b / 2→c / 3→d    D. 1→f / 2→e / 3→d
Evidence: Jigsaw puzzle (12:18:55-13:26:23), corresponding to option f; Whiteboard assembly (12:50:54-13:17:28), corresponding to option c; Second-floor chat (13:17:28-13:26:23), corresponding to option d. The correspondence in option A is correct.

**[Event Context]**
After A6 SHURE completed the jigsaw puzzle, what was the next event that happened?
A. Whiteboard assembly    B. Second-floor chat    C. Takeout lunch ordering    D. Supermarket procurement
Evidence: According to the time sequence, after the jigsaw puzzle (12:18:55-13:26:23) is completed, the next is takeout lunch ordering (13:26:23-14:18:34). The whiteboard assembly (12:50:54-13:17:28) was carried out during the jigsaw puzzle process, the second-floor chat (13:17:28-13:26:23) also occurred during the jigsaw puzzle and before the takeout lunch ordering, and the supermarket procurement (17:18:39-18:37:08) is an event that happens much later. Therefore, the next immediate event is the takeout lunch ordering.

**[Event Rectification]**
A6 SHURE described a series of consecutive actions as: [Whiteboard assembly → Event planning whiteboard meeting → Second-floor chat], but her statement is incorrect. What is the correct sequence and the error analysis?
A. Correct: Event planning whiteboard meeting → Whiteboard assembly → Second-floor chat. Error reason: Incorrect order (event planning whiteboard meeting should precede whiteboard assembly)
B. Correct: Whiteboard assembly → Second-floor chat → Event planning whiteboard meeting. Error reason: Disordered order
C. Correct: Event planning whiteboard meeting → Second-floor chat → Whiteboard assembly. Error reason: Incorrect order
D. Correct: Second-floor chat → Event planning whiteboard meeting → Whiteboard assembly. Error reason: Completely reversed core event order
Evidence: The correct sequence should be Event planning whiteboard meeting (11:23:49-12:18:55) → Whiteboard assembly (12:50:54-13:17:28) → Second-floor chat (13:17:28-13:26:23). A6 SHURE's description reversed the order of "Event planning whiteboard meeting" and "Whiteboard assembly". Option A corrects the core order."

**[Event Reconstruction]**
In A6 SHURE's diary, the record of the order of her participating activities is unclear: [Activity planning whiteboard meeting] → (?) → [Whiteboard assembly] → (?) → [Supermarket procurement]. Which of the following events can be used to fill in the unclear parts?
A. Jigsaw puzzle / Second-floor chat    B. Second-floor chat / Takeout lunch ordering
C. Jigsaw puzzle / Takeout lunch ordering    D. Takeout lunch ordering / Second-floor chat
Evidence: Activity planning whiteboard meeting (11:23:49-12:18:55) → Jigsaw puzzle (12:18:55-13:26:23) → Whiteboard assembly (12:50:54-13:17:28) → Takeout lunch ordering (13:26:23-14:18:34) → Supermarket procurement (17:18:39-18:37:08). The first mask should be filled with "Jigsaw puzzle", which starts immediately after the activity planning whiteboard meeting. The second mask should be filled with "Takeout lunch ordering", which is after the whiteboard assembly and before the supermarket procurement. Option C correctly matches the chronological order.

Table 9: **The manually annotated six event understanding question-answer pairs** for A6 SHURE Day 1 in EgoMem. The correct options are in blue. Each pair is also annotated with corresponding evidence.

**[Detail Perception 1]**
When SHURE attended the whiteboard meeting on Day 1, there were three categories of people written on the whiteboard. The first was the invited guests, the second was the invited neighbors. What was the third category?
A. Teachers         B. Crew members         C. Classmates         D. Colleagues
Evidence: On Day 1, from 11:26:57 to 11:33:15, SHURE wrote three categories of people on the whiteboard. The first was the invited guests, the second was the invited neighbors, and the third was the crew members.

**[Detail Perception 2]**
Before SHURE assembled the whiteboard on Day 1, who used scissors to open the express box containing the whiteboard stand?
A. The woman in black top         B. The boy in blue short-sleeved shirt
C. SHURE         D. The girl in white short-sleeved shirt
Evidence: On Day 1, from 12:50:27 to 12:54:46, SHURE used scissors to open the express box containing the whiteboard stand.

**[Detail Perception 3]**
On Day 1, during the car ride to the supermarket, who sat next to SHURE in the car?
A. The woman in black top         B. The boy in blue short-sleeved shirt
C. The woman in blue dress         D. The girl in white short-sleeved shirt
Evidence: On Day 1, from 17:11:15 to 17:13:04, during the car ride to the supermarket, the boy in blue short-sleeved shirt sat next to SHURE.

**[Detail Perception 4]**
On Day 1, after SHURE finished shopping with the team and took the subway, what was the name of the subway station they walked to?
A. Jiukeshu Station         B. Guoyuan Station         C. Caodi Station         D. Senlin Station
Evidence: On Day 1, from 18:37:23 to 18:45:22, SHURE walked to Jiukeshu Subway Station.

**[Detail Perception 5]**
On Day 1, when SHURE went to the classroom to find his girlfriend, which row of the classroom was their seat in?
A. The first row         B. The second row         C. The third row         D. The fourth row
Evidence: On Day 1, from 20:05:35 to 20:07:10, SHURE entered the classroom, found his girlfriend in the fourth row and took a seat.

**[Detail Perception 6]**
On the evening of Day 1, when SHURE went to have dinner with his girlfriend, what was the number on the waiting sign placed on the table while waiting for the food after ordering?
A. 3         B. 4         C. 43         D. 34
Evidence: On Day 1, from 21:13:22 to 21:20:12, the number on the waiting sign on the table while waiting for the food after ordering dinner was 43.

Table 10: **The manually annotated six detail perception question-answer pairs** for A6 SHURE Day 1 in EgoMem. The correct options are in blue. Each pair is also annotated with corresponding evidence.

---

Please complete the following task:
Carefully analyze the given Text and Question.
Determine whether the Text contains descriptions or information relevant to the Question. Score the relevance on a scale of 1 to 5 based on the following criteria:
1 point: The Text has no relevance to the Question, and there is no content related to the Question in the Text at all.
2 points: The Text has a very weak relevance to the Question, with only a minimal amount of unrelated indirect descriptions.
3 points: The Text has some relevance to the Question, containing some relevant information, but it is not comprehensive or in - depth enough.
4 points: The Text has a relatively strong relevance to the Question, containing a substantial amount of relevant information and being able to respond to the Question fairly well.
5 points: The Text is highly relevant to the Question, fully and thoroughly covering all the key information required by the Question.
Output the final score in the format of "Scoring result: X points", where X is an integer between 1 and 5.
Given Text: {*Text*}
Given Question: {*Question*}

---

Table 12: **The designed prompts for the relevance evaluator**.

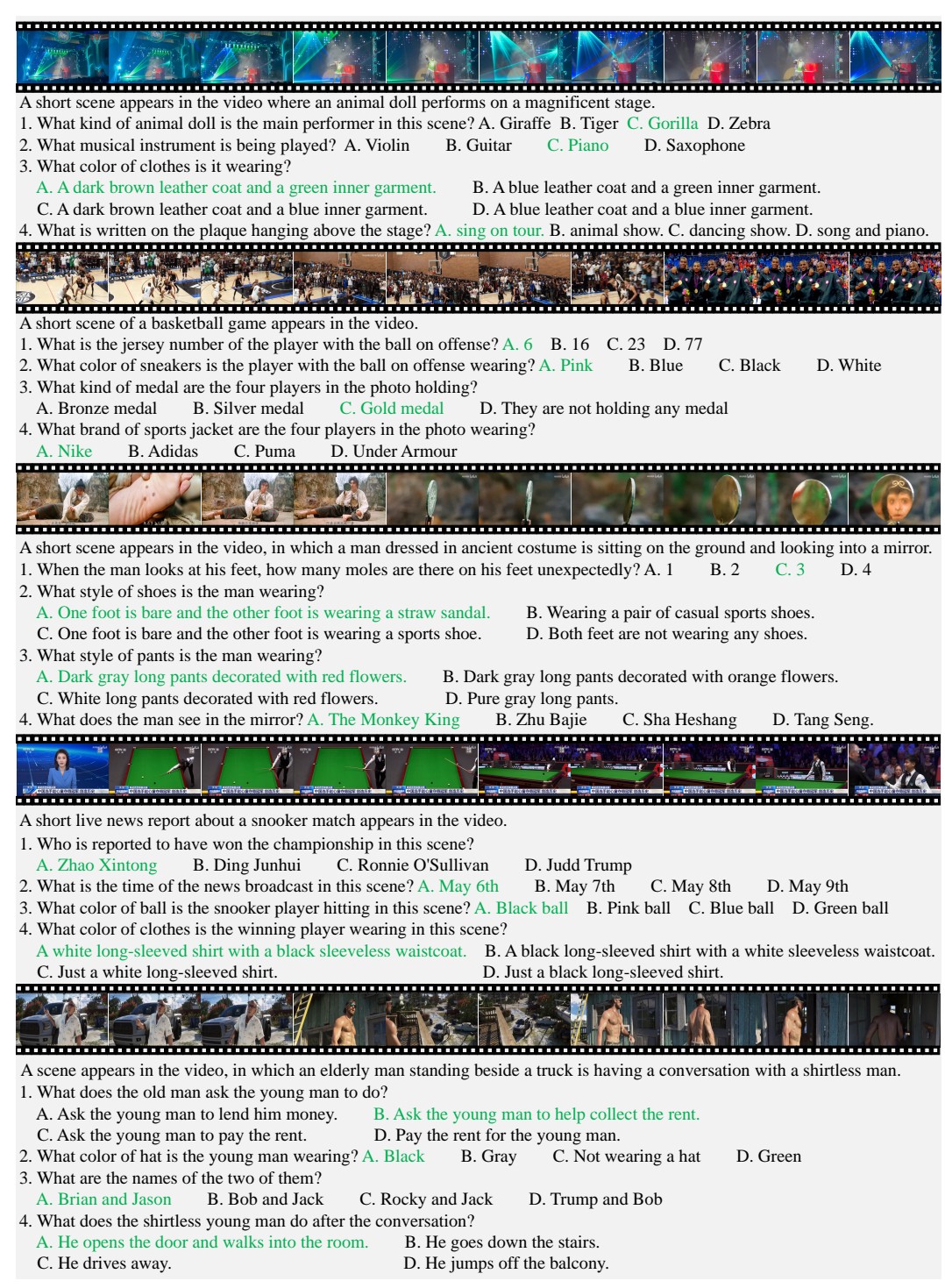

A short scene appears in the video where an animal doll performs on a magnificent stage.
1. What kind of animal doll is the main performer in this scene? A. Giraffe B. Tiger C. Gorilla D. Zebra
2. What musical instrument is being played? A. Violin    B. Guitar    C. Piano    D. Saxophone
3. What color of clothes is it wearing?
   A. A dark brown leather coat and a green inner garment.    B. A blue leather coat and a green inner garment.
   C. A dark brown leather coat and a blue inner garment.    D. A blue leather coat and a blue inner garment.
4. What is written on the plaque hanging above the stage? A. sing on tour. B. animal show. C. dancing show. D. song and piano.

A short scene of a basketball game appears in the video.
1. What is the jersey number of the player with the ball on offense? A. 6   B. 16   C. 23   D. 77
2. What color of sneakers is the player with the ball on offense wearing? A. Pink    B. Blue    C. Black    D. White
3. What kind of medal are the four players in the photo holding?
   A. Bronze medal    B. Silver medal    C. Gold medal    D. They are not holding any medal
4. What brand of sports jacket are the four players in the photo wearing?
   A. Nike    B. Adidas    C. Puma    D. Under Armour

A short scene appears in the video, in which a man dressed in ancient costume is sitting on the ground and looking into a mirror.
1. When the man looks at his feet, how many moles are there on his feet unexpectedly? A. 1    B. 2    C. 3    D. 4
2. What style of shoes is the man wearing?
   A. One foot is bare and the other foot is wearing a straw sandal.    B. Wearing a pair of casual sports shoes.
   C. One foot is bare and the other foot is wearing a sports shoe.    D. Both feet are not wearing any shoes.
3. What style of pants is the man wearing?
   A. Dark gray long pants decorated with red flowers.    B. Dark gray long pants decorated with orange flowers.
   C. White long pants decorated with red flowers.    D. Pure gray long pants.
4. What does the man see in the mirror? A. The Monkey King    B. Zhu Bajie    C. Sha Heshang    D. Tang Seng.

A short live news report about a snooker match appears in the video.
1. Who is reported to have won the championship in this scene?
   A. Zhao Xintong    B. Ding Junhui    C. Ronnie O'Sullivan    D. Judd Trump
2. What is the time of the news broadcast in this scene? A. May 6th    B. May 7th    C. May 8th    D. May 9th
3. What color of ball is the snooker player hitting in this scene? A. Black ball   B. Pink ball   C. Blue ball   D. Green ball
4. What color of clothes is the winning player wearing in this scene?
   A white long-sleeved shirt with a black sleeveless waistcoat.    B. A black long-sleeved shirt with a white sleeveless waistcoat.
   C. Just a white long-sleeved shirt.    D. Just a black long-sleeved shirt.

A scene appears in the video, in which an elderly man standing beside a truck is having a conversation with a shirtless man.
1. What does the old man ask the young man to do?
   A. Ask the young man to lend him money.    B. Ask the young man to help collect the rent.
   C. Ask the young man to pay the rent.    D. Pay the rent for the young man.
2. What color of hat is the young man wearing? A. Black    B. Gray    C. Not wearing a hat    D. Green
3. What are the names of the two of them?
   A. Brian and Jason    B. Bob and Jack    C. Rocky and Jack    D. Trump and Bob
4. What does the shirtless young man do after the conversation?
   A. He opens the door and walks into the room.    B. He goes down the stairs.
   C. He drives away.    D. He jumps off the balcony.

Figure 9: **Display of five short videos used as "needles".** Each short video contains four manually annotated question-answer pairs about detail perception, with the correct options marked in green.

