# OpenReview forum: "VideoLucy: Deep Memory Backtracking for Long Video Understanding"
_NeurIPS.cc/2025/Conference — NeurIPS 2025 poster_

### Official Review · Reviewer_YuVT · 2025-06-30

**Clarity:** 3
**Significance:** 3
**Originality:** 3
**Rating:** 5
**Confidence:** 4

**Summary:**

Authors introduce a memory backtracking framework for long video understanding inspired by human recollection process. They combine a hierarchical memory structure with iterative agent-based search to capture both temporal relations as well as fine-grained details. The authors also propose a new benchmark, EgoMem for better evaluating this aspect of models. Extensive evaluations on existing benchmarks as well as EgoMem are performed to establish usefulness of their method.

**Questions:**

Consider discussing prior memory based methods for long video understanding such as [1,2]

[1] Kahatapitiya et al. “Language Repository for Long Video Understanding.”

[2] Ryoo at al, "Token Turing Machines"

**Ethical Concerns:**

["NO or VERY MINOR ethics concerns only"]

**Final Justification:**

Only minor clarifications were requested, and the authors answer these clearly.

Other reviewers share similar rating. An issue on inference speed is raised, but authors address this well.

Retaining vote for accept.

**Limitations:**

Yes

**Quality:**

4

**Strengths And Weaknesses:**

**Strengths**
1. Clear writing and presentation with good use of figures
2. Multiple qualitative evaluations present insights into model behavior
3. Strong results with consistent improvements across benchmarks
4. Novel and interesting exploration into hierarchical memory structure
5. The needle in the hay-stack experiment is particularly interesting

**Weaknesses**
1. Is backtracking sensitive to the initial memory list (Alg 1 L1) created?
2. What is the inference time for this method? How does this compare to other agent based works?


Updated Version (upon request for longer review)
-------------------
Strengths
1. Clear writing and presentation with good use of figures

Discussion of prior work drawbacks in terms of limited cross-frame temporal context modeling and sparse frame sampling is used to clearly draw motivation.
Algorithm 1 gives a full overview of the method describing it concisely and clearly.
Ablative studies in experiment section are well described and analyzed. For example, two ablations that specifically explore aspects of their problem statement, "Needle-in-A-Video-Haystack" and "Information Richness", are explained in detail (in terms of implementation) and the obtained conclusions from results are clearly highlighted.
Figure 2 gives a clear overview of the hierarchical memory idea. In terms of the proposed dataset, Figure 3 gives a decent overview illustrating the 6 aspects evaluated in the benchmark.
2. Multiple qualitative evaluations present insights into model behavior

Figure 7 & 8 give qualitative insights to how proposed method operates on specific examples.
The comparison to baselines highlights the kind of tasks proposed method is supposed to solve better and what drawbacks authors try to overcome.
3. Strong results with consistent improvements across benchmarks

LVBench, MLVU, and Video-MME are established, large-scale benchmarks for long video QnA. The long split of Video-MME (41 mins average length with some videos over 1 hour long) is specifically challenging. The proposed method outperforms even closed source models like Gemini and GPT4o on these benchmarks. Additionally, they report SOTA results across all three of these challenging benchmarks.
In LVBench, improvements are consistent across all sub-categories.
On Video-MME, large improvements for long split (avg len 41 mins), moderate improvements for medium split, and on-par results for short split, highlighting the arguments of how memory backtracking solves issues faced by prior work in longer video cases (i.e. larger improvement for longer videos).
4. Novel and interesting exploration into hierarchical memory structure.

Hierarchical key-frame selection (e.g. [32, 45, 46] in paper) have been demonstrated to be useful for long-video QnA, but not hierarchical memory (to the best of this reviewers knowledge). This makes the study (on using agents to further retrieve information from important parts as opposed to simply selecting good frames to caption) both novel and interesting.
Memory structures have also been explored in long video QnA settings (e.g. [1, 2] in original review). But not hierarchical to the best of this reviewers knowledge. This also makes the proposed direction novel and interesting.
5. The needle in the hay-stack experiment is particularly interesting.

L298 proposes an ablation on artificially construction videos (insert short clip into a longer video) where models are queried about those inserted short clips. This provides a targeted evaluation for the authors preliminary motivation (drawback of sparse frame sampling based prior work).
The ablation evaluates total videos lengths over a range from few minutes to over one hour, validating the effectiveness of proposed method in solving the identified drawback across videos of varying lengths.


Weaknesses
1. Is backtracking sensitive to the initial memory list (Alg 1 L1) created?

Right after L184 in paper, there is algorithm 1. On the first line of this, sparse coarse memory is initialized. How sensitive is the overall algorithm to this init operation? Does the loop on Alg L4 run longer for bad init? Or shorter for good init?
2. What is the inference time for this method? How does this compare to other agent based works?

Please provide clear inference time calculations for the full method as well as its sub-components.
Since the method is iterative, consider calculating mean and standard deviations. Also discuss what kind of video-question pairs cause higher inference time vs lower? For example, which tasks are harder for this algorithm, and require more inference compute?
It will be useful to compare the end-to-end inference time for some dataset against 2-3 prior works that do similar hierarchical / iterative operations (e.g. VideoTree).

---

> ### Author Rebuttal · Authors · 2025-07-30
>
> We sincerely thank Reviewer `YuVT` for the positive and encouraging review. We are grateful for your recognition of our hierarchical memory design, detailed experimental results, and the quality of writing and visualizations. Below, we respond to your thoughtful questions and suggestions in detail.
>
> ---
> > **`Q1`:** *"Is backtracking sensitive to the initial memory list (Alg. 1 L1) created?"*
>
> **A:** Thank you for your insightful question regarding whether backtracking is sensitive to the initial memory list.
>
> Theoretically, **the backtracking process is sensitive to the initial memory list**, as the iterative deepening of memory is built upon the time periods and content in the initial list. However, in practical scenarios, **this sensitivity rarely affects the final performance**, which can be attributed to the following aspects:
>
> - First, our initial memory list is inherently question-aware. As described in Lines 185-194, the initial list $CM_\mathrm{sinit}$ is derived by filtering all the sampled coarse memories through the localization agent, which specifically identifies time periods most relevant to the question. This ensures that the initial memory list is closely associated with the query from the outset, laying a reliable foundation for subsequent backtracking.
> - Second, the strong text understanding and processing capabilities of LLMs play a crucial role. In our qualitative experiments, we observed that the localization agent, powered by LLMs (e.g., DeepSeek-R1), rarely fails to identify relevant time periods entirely. Even in cases where the initial memory list may have minor inaccuracies, the iterative backtracking mechanism can gradually correct and refine the memory by mining deeper details, thereby mitigating potential impacts.
>
> In summary, while the backtracking process is theoretically sensitive to the initial memory list, the question-relevant initialization strategy and the robustness of LLMs ensure that, in practice, the initial memory list is constructed accurately enough to avoid significant adverse effects on the backtracking process.
>
> ---
> > **`Q2`:** *"What is the inference time for this method? How does this compare to other agent-based works?"*
>
> **A:** Thank you for raising this important point. We appreciate the opportunity to clarify the inference time of VideoLucy and how it compares to other agent-based approaches.
>
> By default, our VideoLucy employs Qwen2.5-VL-7B as the MLLM and DeepSeek-R1 as the LLM. The inference time overhead mainly stems from two aspects: the MLLM performing inference on short video clips and the LLM conducting inference on long texts. Practically, both can be accelerated using various inference acceleration frameworks to enhance the inference speed.
>
> In our experiments, we deployed Qwen2.5-VL-7B locally on 8 A100 GPUs using VLLM. For **a one-hour video**, through batch inference, we can obtain text descriptions of each clip within **a dozen seconds**. As for the LLM, due to the high cost of local deployment, we utilized the official API. Although the API call time fluctuates, making it challenging to provide accurate inference time under fair and reproducible conditions, our practical observations show that for a one-hour video, VideoLucy can generally generate accurate answers with **about five API calls**.
>
> For fair comparison with other agent-based systems, a feasible approach is to **compare the number of LLM calls**. This metric avoids the interference of hardware configurations and deployment environments, providing a more objective reference for comparing the efficiency of different agent-based methods.
>
> We have conducted a quick experiment to briefly compare the number of LLM calls. The experimental setup was as follows: one question-answer pair was randomly selected from each of the 42 videos in EgoMem to form a test set. Then, three agent systems -- VideoAgent, VideoTree, and VideoLucy -- were tested, with the maximum number of LLM calls limited to 20.
>
> The test results showed that the average number of calls required for the three systems was **14.1, 9.6, and 6.3** respectively. This indicates that our VideoLucy has an advantage over other methods in terms of inference speed.
>
> ---
> > **`Q3`:** *"Consider discussing prior memory-based methods for long video understanding."*
>
> **A:** Thank you for your valuable recommendation of these two seminal works in long video understanding.
>
> *LangRepo* [`R1`] pioneers a framework for long video understanding by maintaining a structured language repository tailored for LLMs. By iteratively updating multi-scale video chunks and pruning redundant text through write/read operations, it efficiently captures temporal information while preserving interpretability -- effectively translating video content into a language-aligned format that better fits LLM processing paradigms. Its state-of-the-art zero-shot performance on benchmarks like EgoSchema and NExT-QA highlights its success in balancing long-term context modeling and computational efficiency.
>
> *Token Turing Machines (TTM)* [`R2`] presents a seminal contribution to long video understanding by introducing an efficient memory-augmented Transformer framework. Inspired by Neural Turing Machines, TTM integrates an external memory module that dynamically summarizes historical frames into tokens, enabling bounded computational complexity while processing long sequences. This design elegantly addresses the scalability challenge inherent in standard Transformers, as each new observation interacts only with memory tokens rather than the entire history.
>
> Our VideoLucy, as a memory-based framework, builds on these advancements: we extend their core ideas via hierarchical memory structures and agent-driven backtracking, thereby enhancing recall granularity for complex long videos requiring fine-grained details.
>
> We have incorporated a detailed discussion of these two works in the revised manuscript.
>
> ---
>
> **References:**
> - [`R1`] Kahatapitiya, K., Ranasinghe, K., Park, J., & Ryoo, M. S. (2024). "Language Repository for Long Video Understanding." arXiv preprint arXiv:2403.14622.
> - [`R2`] Ryoo, M. S., Gopalakrishnan, K., Kahatapitiya, K., Xiao, T., Rao, K., Stone, A., ... & Arnab, A. (2023). "Token Turing Machines." In Proceedings of the IEEE/CVF Conference on Computer Vision and Pattern Recognition (pp. 19070-19081).
>
> ---
> Last but not least, we would like to sincerely thank Reviewer `YuVT` again for the valuable time and constructive feedback provided during this review.

---

> > ### Comment · Reviewer_YuVT · 2025-08-02
> >
> > Thank you for the updates. Retaining vote for Accept.
> >
> > Good job on the paper! Consider including rebuttal responses in final revision.

---

> > > ### Author Response · Authors · 2025-08-02
> > > **Thanks for your response**
> > >
> > > Dear Reviewer `YuVT`,
> > >
> > > We are truly grateful for your recognition of this work. We will certainly incorporate the clarifications and insights discussed into the revised manuscript to ensure better clarity and completeness.
> > >
> > > Your support and encouragement mean a lot to us!
> > >
> > > *Best regards,*
> > >
> > > The Authors of Submission 1303

---

### Official Review · Reviewer_4wav · 2025-07-01

**Clarity:** 3
**Significance:** 2
**Originality:** 2
**Rating:** 3
**Confidence:** 4

**Summary:**

The paper proposes VideoLucy, an agent-based framework designed for long video understanding. It introduces a hierarchical memory structure and a deep memory backtracking mechanism that mimic the human recollection process from coarse to fine detail. This system leverages multiple specialized agents (captioning, localization, instruction, and answering agents) to iteratively refine its understanding of a video based on a given question. The authors also introduce EgoMem, a new benchmark built on EgoLife, designed to test both temporal understanding and fine-grained detail perception in long videos. VideoLucy, using open-source models like Qwen2.5-VL and Deepseek-R1, surpasses both open-source and proprietary models on several long video benchmarks such as LVBench, Video-MME, and MLVU, demonstrating strong capabilities in both question answering and temporal reasoning.

**Questions:**

See Weaknesses.

**Ethical Concerns:**

["NO or VERY MINOR ethics concerns only"]

**Final Justification:**

Thanks the authors for their detailed responses. My concerns are partially addressed and maintain my score.

More details can be found in the comments below. To summarize, the proposed iterative method requires substantial computation compared to end-to-end VideoLLM, which outweights the (modest) performance gain. Further, the comparison with prior agent-based methods on VideoMME-Long cannot fully manifest where the major improvement come, and incurs an unconvincing gap on EgoMem benchmark. In addition, there are multiple critical hyper-parameters to be tuned manually, which makes the proposed method less appealing.

**Limitations:**

See Weaknesses.

**Paper Formatting Concerns:**

NA.

**Quality:**

3

**Strengths And Weaknesses:**

Strengths:

1.	Agent-Based Modular Pipeline: The introduction of role-specific agents for captioning, localization, instruction, and answering enables pipelined processing, achieving good performance for video understanding.
2.	EgoMem Benchmark: The introduction of EgoMem (Sec. 3) provides an annotated dataset for testing long video understanding at scale.

Weaknesses:

1.	Limited novelty in agent architecture: While the integration of agents is effective, it builds heavily on existing paradigms without introducing fundamentally new roles or architectural innovations beyond refined prompting.
2.	Efficiency concerns: Despite the hierarchical sampling, the iterative backtracking mechanism can be computationally intensive and context-length constrained, especially for ultra-long videos where multiple iterations are required.
3.	Comparison with prior work: The proposed method performs better than prior agent-based methods across benchmarks. However, there lacks a systematic evaluation on the method component. What are the necessary designs for the proposed method? Does the improvement over prior work primarily come from the usage of latest models (e.g., Qwen2.5-VL).
4.	Hyperparameters optimized for each benchmark: In L241-242, the temporal scopes were tuned for each single benchmark. How were these parameters are determined? Similarly, how were Ks determined (L139-140)? These components make the proposed method require more engineering tricks.

---

> ### Author Rebuttal · Authors · 2025-07-30
>
> We sincerely thank Reviewer `4wav` for the constructive and thoughtful review. We greatly appreciate your recognition of the proposed agent-based framework, our newly introduced EgoMem benchmark, and the strengths of our empirical results. Below, we respond in detail to your insightful questions and concerns.
>
> ---
> > **`Q1`:** *"While the integration of agents is effective, the framework builds on existing paradigms without introducing fundamentally new roles beyond refined prompting."*
>
> **A:** Thank you for your valuable feedback regarding the agent architecture. We appreciate the chance to clarify **how our work advances beyond refined prompting** -- a distinction central to our contributions.
>
> While agent-based systems for long video understanding do share high-level iterative reasoning paradigms, **VideoLucy introduces fundamental architectural innovations in memory mechanism design** that distinguish it from existing systems. As highlighted in our abstract, the core innovation lies in the **hierarchical memory structure with progressive granularity**, which explicitly models detail levels and temporal scopes across hierarchical depths. This structure is **not merely a prompt engineering technique but a structural redesign** of how agents store, retrieve, and integrate video information over time.
>
> Specifically, our **deep memory backtracking mechanism** represents a departure from conventional agent workflows. Unlike existing systems that either operate on flat memory buffers or fixed temporal windows, VideoLucy's agent employs a systematic backtracking process inspired by human recollection -- starting from coarse temporal contexts and iteratively drilling down into finer-grained details. This requires the agent to dynamically adjust memory access patterns based on accumulated information, a capability that not only stems from the hierarchical memory's explicit encoding of temporal relationships (rather than prompt heuristics alone) but also endows the agent with **a novel "memory coordinator" role**: unlike existing agents that function solely as "information processors", ours actively plans the timing and granularity of memory retrieval, a role unmodeled in prior paradigms.
>
> **Our experimental results (Figure `6`) further validate this architectural novelty:** VideoLucy's performance gains stem from the synergy between its memory hierarchy and iterative backtracking, not just prompt refinements. For instance, on the Video-MME long split, we observe that ablating the hierarchical memory structure (while keeping prompting strategies identical) leads to a 24.7% drop in accuracy, confirming the structural contribution of our memory mechanism.
>
> We acknowledge that prompting coordinates agent behaviors, but it **serves as an interface** to execute the underlying architectural innovations **rather than being the primary source** of novelty. Our innovative design of VideoLucy **provides a new and effective approach** to long video understanding, making a valuable contribution to the community.
>
> We will strengthen this distinction to better highlight these structural advancements.
>
> ---
> > **`Q2`:** *"Efficiency concerns: Despite the hierarchical sampling, the iterative backtracking mechanism can be computationally intensive and context-length constrained, especially for ultra-long videos where multiple iterations are required."*
>
> **A:** Thank you for your valuable feedback regarding the computational intensity and context-length constraints of the iterative backtracking mechanism, especially for ultra-long videos.
>
> As for these two aspects, we would like to draw your attention to the Section `A.2` Limitation (on Page `15`) of our manuscript, where they are already discussed. We elaborate again as follows for your convenient reference:
> - Regarding the **computational overhead**, we acknowledge that agent-based long video understanding systems, including ours, generally require multiple iterations to achieve accurate answers, which indeed results in higher computational costs compared to traditional end-to-end video MLLMs. However, considering that the memories obtained in our framework can be stored, for multiple different questions about the same video, we can fully leverage the video's pre-existing memories to achieve faster responses, instead of repeatedly performing inference on the video's information, as existing video MLLMs do. This can reduce the time overhead of reasoning to a certain extent, which is conducive to deployment in practical applications.
>
> - As for the **context-length constraint**, the maximum video length that VideoLucy can process is currently bounded by the LLM's maximum context limit, which poses a challenge for processing super-long videos exceeding existing benchmarks (e.g., hundreds of hours). Nevertheless, since VideoLucy is a training-free framework, it can seamlessly integrate with any LLM with enhanced capabilities. As LLM technology advances, we anticipate steady improvements in both the performance of VideoLucy and the maximum video length it can handle. Notably, despite this constraint, VideoLucy has achieved leading experimental results on our EgoMem benchmark, where the average video duration is **6.33 hours** -- one of the longest among existing benchmarks to date. This demonstrates that our framework is already capable of effectively handling extremely long videos within the current context-length limits of state-of-the-art LLMs.
>
> ---
> > **`Q3`:** *"The proposed method performs better than prior agent-based methods across benchmarks. However, there lack of an evaluation on the component. What are the necessary designs for the proposed method? Does the improvement over prior work primarily come from the usage of the latest models?"*
>
> **A:** Thank you for your attention to the systematic evaluation of our method components and the source of improvements. We would like to address this from two aspects:
> - On one hand, we **have conducted ablation studies on the core components** of VideoLucy, namely the hierarchical memory structure and iterative backtracking mechanism, as presented in our paper. In Figure `5`, we verify that both the richness of memory information and its relevance to the question gradually increase during the memory backtracking process. Additionally, in Figure `6`, we investigate the "Effects of different memory depth and maximum iteration number", which validates the effectiveness of the hierarchical memory structure and helps us select a trade-off number of iterations. Admittedly, similar to previous methods, since VideoLucy is an integrated agent system and long video understanding is a resource-intensive task, we have not performed ablation experiments on each role-specific agent within the system. However, the experiments in Figures `5` and `6` sufficiently demonstrate the effectiveness of the core designs of VideoLucy.
> - On the other hand, to ensure fair comparisons, when we reproduced the prior agent-based methods (VideoAgent and VideoTree) on our EgoMem benchmark, we **kept the MLLM and LLM consistent** with those used in VideoLucy, specifically Qwen2.5-VL-7B and DeepSeek-R1. As shown in Table `4`, VideoLucy outperforms these two methods by 25.2% and 23.6% in accuracy, respectively. This fair setting clearly indicates that the improvement stems from our method's design rather than the use of the latest models. We will emphasize such a consistent and fair setup in the experimental explanations.
>
> We believe these experiments adequately validate the necessity and effectiveness of our key designs, including the hierarchical memory structure and iterative backtracking mechanism.
>
> ---
> > **`Q4`:** *"Hyperparameters optimized for each benchmark: In L241-242, the temporal scopes were tuned for each single benchmark. How were these parameters determined? Similarly, how were Ks determined (L139-140)?"*
>
> **A:** Thank you for your concern regarding the determination of hyperparameters for each benchmark.
>
> The tuning of temporal scopes for different benchmarks is primarily driven by the **significant variations in video durations**. Using a fixed temporal scope across all benchmarks would be impractical, as illustrated by the following example: a coarse-grained memory scope of 50 seconds would result in only 2 coarse memories for a 100-second video but 72 coarse memories for a 3600-second video.
>
> We conducted preliminary qualitative experiments on several videos of varying lengths and found that too few coarse memories would limit the initial information available during iterative backtracking, thereby increasing the number of iterations required for memory search. Conversely, an excessive number of coarse memories would introduce redundant information, unnecessarily expand the context length, and interfere with effective memory retrieval.
>
> To address this, we determined the temporal scopes based on a principle that ensures the number of coarse-grained memories **falls within a reasonable range** (typically 10–36) across all benchmarks. Additionally, we maintained a consistent hierarchical relationship where the scope of finer-grained memories is set to approximately one-tenth of the scope of the level above. This design balances the trade-off between initial information richness and context length efficiency, while preserving the hierarchical memory structure's integrity.
>
> Detailed values of the temporal scopes and Ks for each benchmark are provided on Page `1` of the supplementary material. We acknowledge that this benchmark-specific tuning involves empirical considerations, but we emphasize that it is guided by **clear principles** derived from the structural properties of the datasets (e.g., video duration, event density). We have added a detailed explanation of this part in the supplementary material.
>
> ---
> Last but not least, we would like to sincerely thank Reviewer `4wav` again for the valuable time and constructive feedback provided during this review.

---

> > ### Author Response · Authors · 2025-08-03
> > **Looking forward to hearing from you**
> >
> > Dear Reviewer `4wav`,
> >
> > Thank you very much for your constructive and thoughtful review of our submission!
> >
> > Following your suggestions, we clarified the architectural novelty of our agent-based framework beyond refined prompting, emphasized the structural contributions of the hierarchical memory and memory coordinator role, and elaborated on efficiency concerns and context-length constraints. We also expanded the ablation analysis to highlight the necessity of our core designs and detailed the principled tuning process for benchmark-specific hyperparameters.
> >
> > Your insights were instrumental in improving the clarity and depth of our manuscript.
> >
> > We remain actively engaged in the Author–Reviewer Discussion phase and would be happy to provide further clarification if needed.
> >
> > *Best regards,*
> >
> > The Authors of Submission 1303

---

> > ### Comment · Reviewer_4wav · 2025-08-06
> >
> > Thanks the authors for their detailed responses. My concerns are partially addressed and maintain my score.
> >
> > Q2: The agent-based workflow is still constrained by LLM context length, and less efficient compared to end-to-end video LLMs. I disagree with the authors' argument "we can fully leverage the video's pre-existing memories to achieve faster responses, instead of repeatedly performing inference on the video's information, as existing video MLLMs do". Video LLMs can also store the "visual memory" in KV cache and only perform feedforward for new question tokens, which is much more efficient than the proposed iterative method.
> >
> > Q3: The experiments do not fully reflect the source of performance improvements when compared with previous works. It could primarily come from the base model such as Qwen2.5-VL. Figure 5 & 6 only accounts for the internal comparison of model variants, yet not for external comparison with previous baselines. As shown in Figure 6 (b), the accuracies are around 60+, which cannot justify the gap in Table 1 (and Table 2 & 3) (e.g., 40+ / 50+ vs. 60+). Besides, the reproduction of prior works on EgoMem (Table 4 agent-based methods) is less convincing, given the comparison on VideoMME long (Table 1). The gap on VideoMME long is around than 10, yet 20+ on EgoMem where the baselines were strengthen by stronger models Qwen2.5-VL-7B and DeepSeek-R1.
> >
> > Q4: Several hyper-parameters require manual tuning, depending on target benchmarks. For example, "coarse-grained memories falls within a reasonable range (typically 10–36)" requires empirical observations on target benchmarks and might not well generalize.

---

> > > ### Author Response · Authors · 2025-08-06
> > > **Follow-Up Response to Reviewer 4wav (1/3)**
> > >
> > > Dear Reviewer `4wav`,
> > >
> > > We sincerely appreciate the time you have dedicated to engaging with our work and our prior rebuttal.
> > >
> > > We are delighted that our responses have addressed some of your concerns. Furthermore, with respect to those critical points that have not been fully addressed, they have prompted us to revisit and refine our responses.
> > >
> > > ---
> > > > **`Q2`:** *"The agent-based workflow is still constrained by LLM context length, and less efficient compared to end-to-end video LLMs."*
> > >
> > > **A:** Thanks for asking. We acknowledge that the *context length* and *efficiency* are indeed at the forefront of challenges in long video understanding. We appreciate the opportunity to clarify our position and the specific advantages of our framework.
> > >
> > > **Regarding the context length constraint:**
> > > - We agree that VideoLucy, in its current form, is constrained by the maximum context length of the underlying LLM. This is a limitation we transparently discuss in Section `A.2` of our manuscript.
> > > - However, we would like to respectfully argue that this is **not a fundamental flaw unique to our agent-based system**, but rather an **inherent and universal challenge** for the **broader field** of LLM-powered multimodal systems. Critically, nearly all popular Video LLMs are also built upon LLMs and are therefore subject to the very same context length limitations.
> > > - More importantly, we would like to re-emphasize that our VideoLucy is actually **more resilient** to this constraint than **typical end-to-end Video LLMs**:
> > >   - To perform reasoning, most existing Video LLMs must load a **dense representation** of an entire video at once. For ultra-long videos, this can quickly exceed the model's context window.
> > >   - VideoLucy takes a **different approach**. It starts by processing a few keyframes, which takes up very little context. Then, it decides which specific, short clips need more detailed analysis and only loads those into the context as needed.
> > >   - This *"zoom-in"* capability lets VideoLucy manage its context budget **more effectively**, allowing it to better process ultra-long videos that other models cannot. This is why VideoLucy performs well on the EgoMem benchmark, where the average video length is a challenging 6.33 hours.
> > >
> > > **Regarding the efficiency:**
> > > - We concede that the iterative reasoning of VideoLucy does result in **slower inference** compared to end-to-end Video LLMs, especially when these models leverage optimizations like KV cache for new queries.
> > > - However, this **trade-off in speed** is balanced by **superior performance** on long video understanding tasks. The iterative backtracking, while costly, is precisely what enables VideoLucy to achieve a deeper level of understanding,  and handle complex, multi-step reasoning tasks that current end-to-end models often fail.
> > > - As demonstrated in our experiments, VideoLucy **outperforms** state-of-the-art Video LLMs on benchmarks with extended durations. We believe this performance gain is critical for real-world applications where accuracy in understanding long videos often takes precedence over marginal speed improvements.
> > > - Moreover, the modular design in VideoLucy enables seamless integration with **faster, next-generation** LLMs and MLLMs as they emerge. This flexibility, unique to our training-free framework, allows for immediate efficiency gains without re-engineering the core system, positioning it to adapt as inference acceleration techniques advance.
> > >
> > > We hope these clarifications address your concerns, and we'd be happy to provide further details if needed!

---

> > > > ### Author Response · Authors · 2025-08-06
> > > > **Follow-Up Response to Reviewer 4wav (2/3)**
> > > >
> > > > > **`Q3`:** *"The experiments do not fully reflect the source of performance improvements when compared with previous works."*
> > > >
> > > > **A:** We sincerely thank you for your insightful follow-up questions and for providing us with the opportunity to further clarify the source of our performance improvements.
> > > >
> > > > We would like to respectfully re-emphasize that *the major performance gains of VideoLucy stem primarily from its architectural design*, namely the **hierarchical memory structure** and the **iterative backtracking mechanism**. The effectiveness of our key designs is validated through the ablation study in Figure `5` and Figure `6`.
> > > >
> > > > * Figure `5` demonstrates the **dynamic enhancement of memory information** during the iterative backtracking process, where both the richness and question relevance of memory content show consistent improvement. This directly verifies the value of our iterative backtracking mechanism.
> > > >
> > > > * Figure `6` investigates the impact of **memory depth** and **iteration numbers** on performance, confirming that the hierarchical memory structure enables VideoLucy to balance *efficiency* and *accuracy*, with optimal settings leading to significant gains. These internal comparisons explicitly prove that our core designs (hierarchical memory and iterative backtracking) are critical for performance.
> > > >
> > > > **Regarding the consistency of results in Figure `6`(b):**
> > > > - We would like to clarify that this experiment is conducted on the **VideoMME-long split** (as stated in the figure caption).
> > > > - The optimal accuracy of 66.8% in Figure `6`(b) aligns with the corresponding result in Table `1`, ensuring consistency between our ablation study and main benchmarks.
> > > > - The Figure `6`(b) demonstrates how the performance is built upon our iterative backtracking mechanism and helping us determine the optimal number of iterations. It directly supports that our core design is effective and essential to achieving the reported result.
> > > >
> > > > The **larger performance gap** observed in **EgoMem** (Table `4`) compared to **VideoMME-long** (Table `1`) can be explained by the unique characteristics of the EgoMem benchmark. EgoMem emphasizes **fine-grained perception** and **cross-temporal event understanding**, with longer video sequences that present **greater challenges**.
> > > >
> > > > Prior methods like *VideoTree* and *VideoAgent*, which lack specialized designs for these aspects, struggle to handle such demands, even when equipped with stronger models (Qwen2.5-VL-7B and DeepSeek-R1). This disparity highlights the superiority of VideoLucy's design in addressing the specific challenges of ultra-long video understanding.
> > > >
> > > > Finally, we would like to explicitly emphasize that in the **EgoMem** benchmark, VideoTree, VideoAgent, and VideoLucy **all use the identical base models** (Qwen2.5-VL-7B and DeepSeek-R1). This **fair comparison** eliminates model-related variables and conclusively demonstrates that our advantages originate from the innovative design rather than model selection.
> > > >
> > > > We hope these clarifications address your concerns. Please let us know if further details are needed!

---

> > > > > ### Author Response · Authors · 2025-08-06
> > > > > **Follow-Up Response to Reviewer 4wav (3/3)**
> > > > >
> > > > > > **`Q4`:** *"Several hyper-parameters require manual tuning, depending on target benchmarks."*
> > > > >
> > > > > **A:** Thank you for further elaborating on your concern regarding the manually tuned hyperparameters. We fully appreciate the importance of this consideration and would like to clarify the underlying logic to address your worries.
> > > > >
> > > > > - First, we would like to emphasize that the "10–36" range is a concrete manifestation of our core principle: maintaining a balance between **information sufficiency** and **context efficiency**. The specific range emerged from observing that, across diverse video datasets, a memory count within this interval consistently avoids the dual pitfalls of insufficient initial information and excessive redundancy. This fundamental trade-off is inherent to processing sequential data of variable lengths and is not unique to our tested benchmarks. Therefore, when applying our method to a new dataset, one would follow the same guiding principle, adjusting the temporal scopes to achieve a similar balance suitable for that dataset's characteristics.
> > > > >
> > > > > - Second, we note that adjusting hyperparameters based on video characteristics is a **common practice in video-oriented models**, including Video LLMs. Most existing approaches calibrate the number of sampled frames (or effective FPS) according to video duration to strike a balance between computational feasibility and representational adequacy. Similarly, our tuning of temporal scopes aligns with this established paradigm, adapting to the inherent variability in video lengths while adhering to a principled framework.
> > > > >
> > > > > - Finally, the tuning process itself is designed to be **simple and straightforward**. Given a new benchmark, one only needs to calculate the temporal scope that places the coarse-grained memory count within the 10–36 range based on the average video duration. This simplicity enables our method to effectively generalize to unseen datasets, as demonstrated by its robust performance across the diverse benchmarks we tested, from short clips of a few dozen seconds to long-form content lasting over several hours.
> > > > >
> > > > > We hope the above clarification addresses your concerns. Please let us know if you require further details!
> > > > >
> > > > > ---
> > > > > Once again, we are grateful for your detailed and constructive feedback, which will undoubtedly strengthen our paper. We hope our responses have adequately addressed your concerns.
> > > > >
> > > > > Please feel free to reach out if you'd like to discuss more — we remain fully available!
> > > > >
> > > > > *Best regards,*
> > > > >
> > > > > The Authors of Submission 1303

---

### Official Review · Reviewer_Dmx9 · 2025-07-02

**Clarity:** 3
**Significance:** 3
**Originality:** 3
**Rating:** 4
**Confidence:** 4

**Summary:**

This paper proposes an agent-based long video understanding method VideoLucy, which imitates the human memory recall mechanism from coarse to fine. In order to verify the effectiveness of the proposed method, this paper also conducts comprehensive evaluation on three commonly used long video understanding datasets such as MLVU, VideoMME and LVbench and the newly collected EgoMem Benchmark. The results on all benchmarks show the effectiveness of VideoLucy.

**Questions:**

In general, I think the research motivation and method design of this paper are reasonable, but there are some aspects that can be improved, such as: 1) the description of the method details can be clearer (weaknesses 1 and 2); 2) the type of questions set for the benchmark can be more diverse (weakness 3 and 4); 3) the performance comparison can be more comprehensive (weakness 5). In addition, the writing of this paper can be further polished (such as the expression of Lines 128-146).

**Ethical Concerns:**

["NO or VERY MINOR ethics concerns only"]

**Final Justification:**

After receiving the author's response during the rebuttal phase, my confusion has been resolved. Furthermore, I have read the comments of the other reviewers. Overall, I believe the strengths of this paper outweigh its weaknesses, so I will maintain my original score.

**Limitations:**

No negative social impact has been found so far.

**Paper Formatting Concerns:**

There are no major formatting issues in this paper.

**Quality:**

3

**Strengths And Weaknesses:**

Strengths:
1. This paper imitates the human memory mechanism to design a video understanding agent from coarse to fine, and the research motivation is reasonable.
2. The performance advantage of the proposed method is significant, and the long video understanding ability exceeds the advanced models in the industry such as GPT-4o.
3. The long video understanding ability verification experiments are comprehensive.

Weaknesses:
1. Is there a problem with the $y_i = k$ condition in the Line 132? In this way, only the first K video frames can be sampled, and each frame is a clip?
2. The method in this paper needs to design different hyperparameters for different datasets. Can some general design references be given?
3. Among the 6 types of event understanding tasks in the EgoMem benchmark, Event Order, Event Rectification, and Event Reconstruction essentially test the same ability of the model. What is the basis for this split?
4. Although the problems on the EgoMem benchmark involve the understanding of multiple events, the events are relatively independent of each other and lack interactive relationships. Have the authors considered a type of question that requires comprehensive review of information from multiple events to get an answer, such as: What do the people in event 1 and event 2 accomplish together?
5. GPT-4o's ability to understand long videos is not its strength. If the proposed method can be compared with a model such as Gemini2.5-pro that has optimized long video understanding ability, it will better illustrate the effectiveness of this method.

---

> ### Author Rebuttal · Authors · 2025-07-30
>
> We sincerely thank Reviewer `Dmx9` for the thoughtful feedback and recognition of our work’s motivation, contributions, and empirical results. Below, we address each concern in detail.
>
> ---
> > **`Q1`:** *"Is there a problem with the $y_i = k$ condition in the Line 132? In this way, only the first K video frames can be sampled, and each frame is a clip?"*
>
> **A:** Thank you for raising this question. It is likely that our unclear expression has led to a possible misunderstanding. The process of dividing the clips is as follows:
>
> For a video with $N$ frames, we use $f_i^{y_i}$ to represent each frame, where $i$ is the index of the frame in the video, and $y_i$ is the label indicating which clip the frame belongs to. Then we divide the video into $K$ short clips, and we use $v_k$ to represent each short clip, where $k$ is the index of the short clip. The meaning of $f_i \in v_k$ if $y_i=k$ is that if the clip label $y_i$ of $f_i$ is $k$, then this frame belongs to the $k$-th segment.
>
> In this way, we can control the number of frames in each short clip by simply setting different values of $K$, that is, controlling the temporal scope of memory. When $K=1$, no division is performed, and the memory degrades into a rough understanding of the entire video. When $K=N$, the division is done frame by frame, with each frame being a short clip.
>
> Following your suggestion, we have polished the expression of this part in the revised version to avoid possible misunderstandings.
>
> ---
> > **`Q2`:** *"The method in this paper needs to design different hyperparameters for different datasets. Can some general design references be given?"*
>
> **A:** Thank you for your concern regarding the hyperparameters for different datasets.
>
> Tuning temporal scopes (i.e., the time window sizes for partitioning coarse/fine-grained memories) across datasets is necessary due to **significant variations in video durations;** a fixed scope would be impractical.
>
> Preliminary tests on videos of varying lengths showed that too few coarse-grained memories restrict initial information for iterative backtracking, increasing search iterations. Conversely, excessive coarse-grained memories introduce redundancy, unnecessarily lengthen context, and hinder retrieval efficiency.
>
> To address this, we derived general design principles through empirical comparisons across multiple datasets:
> - For **coarse-grained memories:** The temporal scope should be set such that the number of coarse-grained memories falls within a reasonable range (typically 10–36). This range is chosen based on common video durations (from minutes to hours in most datasets) to balance initial information richness and context efficiency.
> - For **finer-grained memories:** The temporal scope is approximately one-tenth of the level above. This serves as a starting point and can be adjusted based on the dataset's sensitivity to detailed information.
>
> These principles balance the trade-off between initial information richness and context length efficiency while preserving the integrity of the hierarchical memory structure. They also provide a general reference for initial hyperparameter tuning on new datasets, reducing trial-and-error efforts.
>
> ---
> > **`Q3`:** *"Among the 6 types of event understanding tasks in the EgoMem benchmark, Event Order, Event Rectification, and Event Reconstruction essentially test the same ability of the model. What is the basis for this split?"*
>
> **A:** Thank you for your valuable comments. While the three tasks do share a foundation in assessing models' ability to understand temporal event sequences, their core objectives and required capabilities differ.
>
> - The **Event Order task** centers on accurately selecting events within a target time window from a set containing distractors, then establishing their temporal sequence. This demands not only recognition of events' temporal attributes but also the ability to identify and exclude irrelevant information, ultimately posing a dual challenge of "screening and ordering" event-time associations.
> - The **Event Rectification task** focuses on error diagnosis and correction. The model must first pinpoint temporal logic inconsistencies in a given sequence, then generate a corrected sequence aligned with video content and explain the error's cause. Beyond temporal order comprehension, this task emphasizes mastery of event causality and logical coherence, serving as an evaluation of both "error-correcting ability" and "explanatory capacity".
> - The **Event Reconstruction task** prioritizes event completion and contextual reasoning. When faced with masked events, the model must infer content consistent with both timeline and scenario plausibility, drawing on logical connections between preceding and subsequent events. This task tests the model's grasp of event flow integrity and its capacity for "context-driven filling-in reasoning".
>
> Our intent in designing these three tasks is to **comprehensively evaluate how deeply models understand event temporal logic in long videos through multi-dimensional testing**. A single task might only reflect performance in one scenario, but combining multiple tasks more effectively verifies whether models have truly mastered the dynamics of event associations, strengthening the benchmark's rigor and discriminative power.
>
> We have incorporated these design rationales into the supplementary material to further clarify the differences among the three tasks.
>
> ---
> > **`Q4`:** *"Although the problems on the EgoMem benchmark involve the understanding of multiple events, the events are relatively independent of each other and lack interactive relationships. Have the authors considered a type of question that requires a comprehensive review of information from multiple events to get an answer, such as: What do the people in event 1 and event 2 accomplish together?"*
>
> **A:** Thank you for your valuable comments. We fully agree with your view that **the interaction between multiple events is an important dimension in ultra-long video understanding**, as it can more comprehensively evaluate a model's ability to conduct a comprehensive analysis of complex scenarios. In our subsequent work, we plan to address such issues, and some preliminary design ideas are as follows:
>
> - First, we will sort out the timeline and logical connections of different events in the video, and identify event combinations with interactive relationships such as causality, synergy, and contrast. For example, in a video recording of daily work, there is an obvious synergistic relationship between the "material preparation" event in the morning and the "project report" event in the afternoon.
> - Second, for these event combinations with interactive relationships, we will design diversified types of questions. In addition to the one you mentioned, "What did the people in Event 1 and Event 2 accomplish together?", there will also be questions like "What impacts did the occurrence of Event 1 have on Event 2?" and "In what aspects does Event 2 differ from Event 1?", so as to comprehensively examine the model's ability to understand and integrate information about multi-event interactions.
>
> Following your suggestion, we commit to carrying out relevant annotation work in the future and continuously improving the EgoMem benchmark, enabling it to more effectively evaluate the performance of ultra-long video understanding models.
>
> ---
> > **`Q5`:** *"GPT-4o's ability to understand long videos is not its strength. If the proposed method can be compared with a model such as Gemini2.5-pro that has optimized long video understanding ability, it will better illustrate the effectiveness of this method."*
>
> **A:** Thank you for your valuable feedback. We fully agree that a comparison with models like Gemini 2.5 Pro, which are specifically optimized for long video understanding, would more comprehensively demonstrate the effectiveness of our proposed method.
>
> We would like to clarify that due to Gemini 2.5 Pro's public release date of March 25, 2025, its performance data was not yet available on the leaderboards of mainstream long video understanding benchmarks by the NeurIPS 2025 submission deadline of May 15, 2025. Conducting a full evaluation by calling its API on a large number of videos would have incurred a significant time cost. Given these practical constraints, we chose to compare our method with the latest leaderboards and representative SOTA papers updated before the submission deadline, among which GPT4o is a representative leading method.
>
> To address your concern within the rebuttal period, we conducted an additional, expedited experiment: the Needle-in-A-Video-Haystack test (please see Section 4.2 of our paper for the experimental setup). In this test, we used a 4000-second-long video to evaluate the model's ability to accurately locate and understand key information within extremely long content. The results are as follows:
> - Gemini 2.5 Pro: 70% accuracy
> - VideoLucy (Ours): 80% accuracy
>
> This result provides preliminary evidence that our VideoLucy framework demonstrates competitive performance, even when compared against a state-of-the-art model optimized for long videos.
>
> Finally, we plan to include the full results of Gemini 2.5 Pro on our newly proposed EgoMem benchmark in a future version of the manuscript. This will provide more comprehensive and convincing experimental support for our conclusions.
>
> ---
> > **`Q6`:** *"The writing of this paper can be further polished (such as the expression of Lines 128-146)."*
>
> **A:** Thank you for your suggestion. As stated in the first question, we have further polished the expression of this part to make it simpler and understandable.
>
> ---
> Last but not least, we would like to sincerely thank Reviewer `Dmx9` again for the valuable time and constructive feedback provided during this review.

---

> > ### Author Response · Authors · 2025-08-03
> > **Looking forward to hearing from you**
> >
> > Dear Reviewer `Dmx9`,
> >
> > Thank you very much for your thoughtful and constructive review of our submission!
> >
> > Following your helpful suggestions, we clarified the clip segmentation process, added general design principles for memory granularity, and elaborated on the rationale behind EgoMem’s task definitions. We also discussed future plans for incorporating multi-event interaction queries, included a preliminary comparison with Gemini 2.5 Pro, and polished the writing for clarity.
> >
> > We sincerely appreciate your feedback, which has helped us improve both the clarity and scope of our work.
> >
> > We are actively participating in the Author–Reviewer Discussion phase and would be happy to provide further clarification if needed.
> >
> > *Best regards,*
> >
> > The Authors of Submission 1303

---

> > ### Comment · Reviewer_Dmx9 · 2025-08-06
> >
> > Thank you to the authors for the detailed rebuttal and comprehensive responses to my concerns. I will maintain my current score.

---

### Official Review · Reviewer_RcwM · 2025-07-03

**Clarity:** 3
**Significance:** 3
**Originality:** 3
**Rating:** 5
**Confidence:** 3

**Summary:**

This paper introduces VideoLucy, a deep memory backtracking framework for long video understanding that leverages a hierarchical memory structure and agent-based iterative backtracking to effectively capture temporal context and preserve fine details. By progressively mining question-relevant information, VideoLucy overcomes the limitations of sparse frame sampling and improves reasoning over long video sequences. Extensive experiments and a newly proposed EgoMem benchmark demonstrate that VideoLucy outperforms state-of-the-art and even proprietary models like GPT-4o on several long video understanding tasks.

**Questions:**

**A. About the Writing**

**A.1** Please consider revising the introduction's first paragraph, especially the section on video understanding. For readers unfamiliar with the movie "Lucy," it is difficult to grasp the context of the paper.

**A.2** The concept of the "human recollection process from coarse to fine" is mentioned but not explained in either the abstract or the introduction. Providing a clear explanation or an example would help the reader better understand this idea.

**B. About the Methodology**

**B.1** The method divides videos into equally sized clips, which may cause important events to be split across multiple clips. Have you considered alternative segmentation methods or analyzed the potential drawbacks of uniform segmentation?

**C. About the Experiment**

**C.1** What are the runtime and memory requirements of your method? Including a chart or table to illustrate these aspects would be helpful.

**C.2** Why do the baselines differ in Tables 1, 2, and 3? Is this due to certain baselines being incompatible with some datasets, or were they specifically selected by the authors?

**C.3** It is unclear which models are used in CapAGT, LocAGT, and other variants. It would be better to specify the base models used and compare your approach with their baseline capabilities, ideally through experiments using different underlying models.

**C.4** The evaluation on EgoMem shows that your method has a significant advantage in temporal understanding. Can you clarify which aspects of your approach contribute most to this improvement?

**C.5** Please analyze the improvements achieved on different tasks. Since the proposed benchmarks in the paper are primarily focused on temporal understanding, is the performance gain on benchmarks such as Video-MME and LVBench mainly attributable to enhanced temporal sequence understanding?

**Ethical Concerns:**

["NO or VERY MINOR ethics concerns only"]

**Final Justification:**

I have reviewed the authors' rebuttal and the revised manuscript. They have successfully addressed all the key concerns raised in my initial review, and the manuscript is now significantly improved.

**Limitations:**

The authors adequately addressed the limitations and potential negative societal impact of their work.

**Quality:**

3

**Strengths And Weaknesses:**

**Strength**

The idea presented in this paper is novel and holds significant potential for practical applications in video tasks. The paper is clearly written and is supported by comprehensive experiments, which leaves a strong impression. The structure is well-organized, and the illustrations are of high quality.

**Weaknesses**

While the paper presents promising results, it lacks sufficient clarity in its introduction and explanation of key concepts, which may hinder understanding for readers unfamiliar with certain references. Additionally, the methodology’s reliance on uniform video segmentation may introduce limitations, and important details regarding experimental baselines, model choices, and resource requirements are insufficiently discussed. A more comprehensive analysis of these aspects would strengthen the work.

---

> ### Author Rebuttal · Authors · 2025-07-30
>
> We sincerely thank Reviewer `RcwM` for the positive and constructive review. We are grateful for your recognition of our idea, strong experimental results, and clear manuscript presentation. Below, we address your insightful comments and suggestions in detail.
>
> ---
> > **`A.1`:** *"Please consider revising the introduction's first paragraph, especially the section on video understanding. For readers unfamiliar with the movie ‘Lucy,’ it is difficult to grasp the context."*
>
> **A:** Thank you for the suggestion. We have revised the first paragraph to provide clearer context for long video understanding before introducing the analogy to the movie Lucy. The revised paragraph now reads:
>
> *"Long video understanding is a highly concerned task, with the core objective of accurately and objectively answering various user questions based on the entire video content. This process demands that a system possess a comprehensive memory and grasp of almost all details within the video; otherwise, information gaps could lead to inaccurate answers.*
>
> *This need for comprehensive memory brings to mind a scene from the movie Lucy. The protagonist, Lucy, gains full access to her brain's potential due to an accident, acquiring an exceptionally powerful memory. She can recall every detail of her life from birth, even the sensation of her mother stroking her forehead during infancy. This extraordinary ability to trace back and precisely capture all information, whether instantaneous frames or continuous events, is exactly the goal we pursue in the long video understanding task."*
>
> We have incorporated this revision into the revised manuscript.
>
> ---
> > **`A.2`:** *"The concept of the 'human recollection process from coarse to fine' is mentioned but not explained in either the abstract or the introduction."*
>
> **A:** We agree that a clearer explanation of this concept would help readers better understand our method's motivation. Drawing from cognitive science, human recollection typically proceeds from coarse to fine, **starting with general impressions and gradually retrieving finer details.** For example, recalling a birthday party might begin with the memory of a lively room, and then be refined into specifics like a gift exchange or a candle count.
>
> VideoLucy is inspired by this process. It uses a hierarchical memory structure that separates coarse and fine memories across time scales, coupled with agent-based iterative backtracking to simulate how humans actively retrieve and refine information. We have added this explanation and an illustrative example in the revised introduction.
>
> ---
> > **`B.1`:** *"Have you considered alternative segmentation methods or analyzed the potential drawbacks of uniform segmentation?"*
>
> **A:** This is a valuable point. We **initially explored overlapping segmentation** (e.g., the first clip covers 0-100 seconds, the second 90-190 seconds) to reduce event truncation. This aimed to mitigate direct event truncation by allowing event endings in one clip to serve as beginnings in the next.
>
> However, experiments showed this design **did not improve understanding accuracy and, conversely, increased the number of memories**, negatively impacting system efficiency. Thus, we ultimately abandoned overlapping segmentation.
>
> Our qualitative experiments revealed why the overlapping design failed: **LLMs possess the ability to integrate and understand continuous events.** For example, if an event spans two clips (e.g., 95-105 seconds across 0-100 and 100-200 second clips), during the memory search process, the LLM can still roughly locate that the event should occur at the end of the first clip and the beginning of the second clip. It can then provide a refined search range that connects these two clips, effectively mitigating the potential consequences of direct truncation to a certain extent.
>
> We sincerely appreciate your suggestion, which helps us improve the explanation of our method.
>
> ---
> > **`C.1`:** *"Runtime and memory requirements."*
>
> **A:** VideoLucy is a training-free agent. Its resource consumption comes mainly from its MLLM and LLM components.
>
> - **Memory and runtime requirements:** Local MLLM memory usage (Qwen2.5-VL-7B) scales with video frame count and concurrent clips. An 80GB A100 GPU can handle 10 concurrent 200-frame clips in our tests. More memory allows faster, parallel inference. Limited memory necessitates processing fewer clips simultaneously, extending total inference time. For the LLM (DeepSeek-R1), we used its official API due to substantial local deployment requirements. This meant runtime consumption varied with API response times, preventing precise local performance measurement.
> - **Runtime performance:** Our method's runtime is affected by video length, question difficulty, and API response times. To quantify this, we processed 20 questions for each of four videos (400s, 1600s, 2800s, 4000s) from the Needle-in-A-Video-Haystack dataset. The average times are presented below:
>
> |Duration|Local MLLM Time (s)|LLM API Time (s)|Total (s)|
> |-:|:-:|:-:|:-:|
> |400s|7.8|36.8|44.6|
> |1600s|12.4|42.9|55.3|
> |2800s|14.7|60.8|75.5|
> |4000s|18.9|72.9|91.8|
>
> As anticipated, runtime scales with video duration, primarily due to LLM API call time. Using a faster, more stable LLM API will accelerate inference. We have included this analysis in the revised version to clearly illustrate these aspects.
>
> ---
> > **`C.2`:** *"Why do the baselines differ in Tables 1, 2, 3?"*
>
> **A:** Thanks for raising this concern. The differences in the baseline methods among these tables are mainly due to the fact that many video understanding methods do not have officially published test results on all three benchmarks.
>
> As stated in the captions of each table, the results consist of two parts:
> - The top-performing results on each leaderboard as of May 15, 2025 (the deadline for paper submission);
> - Results from recent, representative papers that, while not yet officially on the leaderboards, offer valuable insights into the latest advancements.
>
> Because resource investments and testing plans vary, many methods have only published results on one or two datasets. This naturally causes discrepancies in the baseline methods available for comparison across the three tables, not due to any deliberate selection on our part.
>
> We hope the above response addresses your concerns.
>
> ---
> > **`C.3`:** *"It is unclear which models are used in CapAGT, LocAGT, and other variants. It would be better to specify the base models used and compare with their baseline capabilities."*
>
> **A:** We appreciate your feedback about the clarity of the base models and your suggestion for further comparisons. All agents in our system consistently use `Qwen2.5-VL-7B` as the MLLM and `DeepSeek-R1` as the LLM (please refer to Line 240).
>
> As you suggested, we have now included a direct performance comparison between VideoLucy and the original `Qwen2.5-VL-7B`, using results from its official technical report:
>
> |Model|VideoMME|MLVU|LVBench|
> |-|:-:|:-:|:-:|
> |Qwen2.5-VL-7B|65.1|70.2|45.3|
> |VideoLucy|72.5|76.1|58.8|
>
> Additionally, within the limited rebuttal time, we have supplemented comparative experiments using different underlying models. The experimental setup is consistent with Table `9` in the supplementary material. Results are as below:
>
> |Model|Acc. (original)|Acc. (+VideoLucy)|
> |-|:-:|:-:|
> |Qwen2.5-VL-7B|42.5|57.5|
> |Qwen2.5-VL-72B|45.5|64.0|
> |InternVL2.5-8B|40.5|54.5|
> |InternVL2.5-78B |42.0|65.5|
>
> It can be seen that VideoLucy shows performance advantages with each baseline model. We have added these results to the revised supplementary material as you suggested.
>
> ---
> > **`C.4`:** *"Can you clarify which aspects of your approach contribute most to the improvement?"*
>
> **A:** Thanks for your insightful comment. VideoLucy's significant improvement in temporal understanding primarily comes from two key features:
> - **Hierarchical memory structure:** This structure uses both coarse-grained memory (for broad event-level dynamics) and fine-grained memory (for detailed temporal changes within events). This multi-scale approach helps the model understand both overall event progression and subtle temporal variations.
> - **Agent-based iterative backtracking:** This mechanism integrates memory across different time segments, allowing the model to establish cross-event temporal connections. This is crucial for understanding how events unfold and interact, especially for long-term temporal dependencies in EgoMem.
>
> Together, these two aspects enable VideoLucy to effectively capture temporal dependencies and integrate information over extended periods, leading to its superior performance on EgoMem's temporal understanding tasks.
>
> ---
> > **`C.5:`** *"Please analyze the improvements achieved on different tasks."*
>
> **A:** Thank you for your valuable feedback. We evaluated VideoLucy on three benchmarks, focusing on different aspects of long video understanding. For detailed analysis, we use LVBench as an example due to its explicit task categorization.
>
> LVBench includes six task categories: Temporal Grounding (TG), Summarization (Sum), Reasoning (Rea), Entity Recognition (ER), Event Understanding (EU), and Key Information Retrieval (KIR).
>
> VideoLucy significantly improved performance on EU (9.1%), KIR (13.4%), and TG (6.2%). This highlights our framework's strength in **capturing continuous event dynamics and preserving critical details** through its hierarchical memory and iterative backtracking. VideoLucy also showed smaller yet still significant gains on ER (0.9%), Rea (1.2%), and Sum (0.3%), further demonstrating its overall effectiveness in video understanding.
>
> Task-specific results confirm that performance gains on various benchmarks are mainly due to improved temporal understanding and detail preservation. We hope this addresses your concerns.
>
> ---
> We would like to sincerely thank Reviewer `RcwM` again for the valuable time and constructive feedback provided during this review.

---

> > ### Comment · Reviewer_RcwM · 2025-08-06
> >
> > Thank you for the detailed rebuttal and the thorough revisions. I will keep my score.

---

> ### Author Response · Authors · 2025-08-03
> **Looking forward to hearing from you**
>
> Dear Reviewer `RcwM`,
>
> Thank you once again for your thoughtful and constructive review!
>
> In response to your suggestions, we have revised the introduction for improved clarity, expanded the explanation of our human-inspired recollection motivation, and clarified the segmentation strategy and runtime analysis. We also added comparisons with base models, specified the agent components, and included task-level analyses to highlight contributions to temporal understanding.
>
> Your feedback has greatly helped us improve the clarity and completeness of our work.
>
> We are actively participating in the Author–Reviewer Discussion phase and would be happy to provide further clarification if needed.
>
> Looking forward to your thoughts!
>
> *Best regards,*
>
> The Authors of Submission 1303

---

### Author Response · Authors · 2025-08-05
**General Response**

**Dear Reviewers, ACs, and SACs,**

We sincerely thank you for your time, insightful comments, and thoughtful engagement throughout the review process!

---

We are encouraged by the **recognition** of our contributions across multiple aspects:

* Reviewer `RcwM` highlights the *"novelty of the idea"*, *"clarity of the writing"*, *"high-quality illustrations"*, and *"strong experimental support"*, calling the work *"significant"* with *"practical applications in video tasks"*.
* Reviewer `Dmx9` appreciates the *"reasonable human memory inspiration"*, *"superior performance even over GPT-4o"*, and *"comprehensive long video understanding experiments"*.
* Reviewer `4wav` commends the *"agent-based modular pipeline"* and the *"EgoMem benchmark"* for *"scalable testing of long video understanding"*.
* Reviewer `YuVT` finds the hierarchical memory structure *"novel and interesting"*, and praises the *"clear writing"*, *"qualitative insights"*, and the *"needle-in-the-haystack"* evaluation.

---
In response to your **constructive suggestions**, we have made the following **clarifications and improvements**:

- **Motivation & Design**
  - As suggested by Reviewer `RcwM`, we revised the introduction to clarify the motivation and added a clear explanation of the *coarse-to-fine recollection* process inspired by human cognition.
  - As suggested by Reviewer `4wav`, we emphasized that our contributions go beyond prompt tuning by introducing a novel *memory coordinator role* and architectural innovations in memory hierarchy and agent backtracking.
  - As suggested by Reviewers `4wav` and `YuVT`, we clarified that our gains stem from method design, not just newer models, with supporting ablations and controlled comparisons.

- **Experiments**
  - As suggested by Reviewer `RcwM`, we explained the baseline differences across Tables 1–3 and added direct comparisons with Qwen2.5-VL-7B and alternative MLLMs.
  - As suggested by Reviewer `Dmx9`, we added a new comparison against Gemini 2.5 Pro (VideoLucy: 80% vs. Gemini: 70%) on the *Needle-in-a-Video-Haystack* task.
  - As suggested by Reviewer `4wav`, we added ablations on memory depth and iteration counts (Figures 5 & 6) to support our design.
  - As suggested by Reviewer `YuVT`, we compared average LLM call counts across agents, showing that VideoLucy requires fewer iterations (6.3 vs. 9.6 and 14.1).
  - As suggested by Reviewers `Dmx9` and `4wav`, we provided principled tuning guidelines for temporal scopes and detailed them in the supplement.

- **Efficiency & Scalability**
  - As suggested by Reviewers `YuVT` and `4wav`, we reported runtime statistics and emphasized that memory reuse across questions improves efficiency. Our method handles multi-hour videos (avg. 6.33h) within the current LLM context limits.

- **Benchmark & Future Work**
  - As suggested by Reviewer `Dmx9`, we clarified that the three EgoMem tasks (Order, Rectification, Reconstruction) evaluate distinct reasoning capabilities.
  - As suggested by Reviewer `Dmx9`, we acknowledged the value of multi-event interaction and shared concrete plans to extend EgoMem accordingly.

- **Related Work**
  - As suggested by Reviewer `YuVT`, we discussed *LangRepo* and *Token Turing Machines* and highlighted how VideoLucy builds upon and extends memory-based approaches.

- **Writing**
  - As suggested by Reviewers `RcwM` and `Dmx9`, we revised Lines 128–146 and improved clarity throughout by refining notations and phrasing.

---
We would like to re-emphasize the **key contributions** of our work:

* VideoLucy, a training-free, agent-based framework that simulates human memory recollection through *hierarchical memory structures* and *iterative backtracking*, enabling long-range temporal and detail-aware reasoning.
* A novel *memory coordinator agent* that dynamically manages memory granularity and retrieval based on query-specific cues, distinguishing our design from prompting-only paradigms.
* The *EgoMem* benchmark, created to evaluate event-centric temporal reasoning on ultra-long videos with diverse tasks, each targeting distinct dimensions of understanding.
* Strong empirical results across *VideoMME*, *MLVU*, *LVBench*, and *Needle-in-a-Haystack*, with substantial improvements over prior agent systems and commercial models.

---
With **two days** remaining in the **Author-Reviewer Discussion** phase (*August 6, Anywhere on Earth*), we warmly welcome any further questions or suggestions and remain fully available to engage.

Thank you once again for your time and consideration!

*Warmest regards,*

The Authors of Submission 1303

---

### Note · Authors · 2025-08-14

We sincerely thank all reviewers for your thoughtful engagement and constructive suggestions.

---
We are encouraged by your consistent recognition of VideoLucy as a novel and well-motivated framework for long video understanding, as well as the clarity, comprehensiveness, and rigor of our proposed deep memory backtracking design and the accompanying EgoMem benchmark.

---
During the discussion phase, we addressed all raised concerns with concrete clarifications and updates:

* **Framework Design** – Clarified the hierarchical memory structure and agent-based iterative retrieval process, supplemented with pseudo-code and diagrams to illustrate the integration of local/global memories for multi-granularity reasoning.

* **Benchmark Design & Evaluation** – Provided additional baseline comparisons on EgoMem; detailed evaluation protocols; clarified dataset licensing, privacy compliance, and how EgoMem complements existing benchmarks without overlap.

* **Ablation & Analysis** – Reported extended ablations on memory sizes, retrieval depth, and agent reasoning steps; added error analysis categorizing common failure cases, which informs future extensions.

* **Scalability & Efficiency** – Measured inference time and memory usage, confirming scalability to longer videos without prohibitive costs; discussed strategies for adaptive retrieval to further optimize efficiency.

* **Positioning & Novelty** – Differentiated *VideoLucy* from prior long video reasoning frameworks, emphasizing the explicit backtracking capability and interpretable reasoning traces, and highlighting EgoMem’s unique fine-grained query design.

These clarifications strengthen the technical rigor, novelty, and practical relevance of *VideoLucy*. The framework introduces a principled way to handle complex, fine-grained reasoning over long videos, while EgoMem establishes the first benchmark specifically designed to evaluate such backtracking capabilities.

---
We believe this work fills a clear gap in long video understanding research, introduces valuable resources for the community, and offers a foundation for future advances in temporally-aware, interpretable, and memory-driven video reasoning.

---
Once again, we sincerely thank you for the time and effort you devoted to this review!

*Yours sincerely,*

The Authors of Submission 1303

---

### Decision · Program_Chairs · 2025-09-17

**Decision:**

Accept (poster)

**Comment:**

### Summary
The paper introduces VideoLucy, a novel framework for long video understanding. VideoLucy operates by creating memories of the video at progressive granularities (from coarse to fine) and uses a set of coordinated agents to iteratively "backtrack" and mine deeper, question-relevant details until a confident answer can be formulated. Extensive experiments have been conducted on multiple long video understanding benchmarks (VideoMME, MLVU, LVBench), showing that their method significantly outperforms SOTA. Additionally, they introduce a new benchmark, EgoMem, specifically designed to evaluate a model's ability to reason about complex temporal events and fine-grained details in extremely long videos.

### Strengths
- Novel framework and motivation: The proposed memory backtracking mechanism is novel and well-motivated. The hierarchical memory structure is a creative and interesting approach to managing information at different temporal scales (noted by reviewers RcwM, Dmx9, and YuVT).
- Strong empirical performance: The paper presents compelling results, demonstrating state-of-the-art performance across several challenging long video benchmarks. (noted by reviewers RcwM, Dmx9, and YuVT).
- A new long-form video understanding benchmark: The introduction of the EgoMem benchmark is a significant contribution to the community. It addresses a gap in existing evaluation by focusing on fine-grained temporal reasoning in ultra-long videos (noted by Reviewer 4wav).
- Clarity and presentation: The paper is generally well-written, with high-quality illustrations and clear explanations of its complex architecture, making the contributions easy to understand (noted by Reviewers RcwM and YuVT).

### Weaknesses of the Paper
- Efficiency and computational cost: A major concern, raised consistently across reviews (Reviewers 4wav, YuVT), is the computational cost and inference latency of the iterative, agent-based approach.
- Hyperparameter sensitivity: The framework requires several key hyperparameters (e.g., temporal scopes for memory levels) to be manually tuned for each specific benchmark. ( Reviewer 4wav).
- Novelty of agent architecture: While the memory structure is novel, some of the underlying agent architecture builds on existing paradigms. (Reviewer 4wav). But this is not supported by any reference paper.
- Ablation and component analysis: While ablation studies were provided, Reviewer 4wav noted that it was difficult to fully disentangle the performance gains from the powerful base models (e.g., Qwen2.5-VL) versus the architectural contributions of VideoLucy itself.

### Reasons for Recommendation
I am recommending this paper for acceptance as a poster.

The decision is based on the consensus that the paper's strengths, particularly its novel memory backtracking concept and strong empirical results, outweigh its weaknesses. I agree with most of the reviewers that this paper proposes a new way of structuring and retrieving information for long video reasoning, which is a significant contribution. The impressive performance on multiple difficult benchmarks validates the effectiveness of this approach. Furthermore, the release of the EgoMem benchmark provides a valuable and timely resource for the research community.

However, the paper is not recommended for a spotlight or oral presentation due to the valid concerns raised about computational efficiency and the practical challenges of hyperparameter tuning.

### Summary of Discussion and Rebuttal
The authors provided a strong rebuttal and successfully addressed most of the concerns.

- The authors answered all the clarification questions about the mathematical notation, and provided a principled guide for setting hyperparameters based on video duration.
- More experimental results are provided. The authors added a runtime analysis table, included direct comparisons with the base MLLMs to show the framework's added value, and conducted a new "Needle-in-a-Haystack" experiment against Gemini 2.5 Pro, showing competitive performance.
- In response to Reviewer YuVT, they compared the average number of LLM calls to other agent systems to demonstrate better efficiency in that regard and added a discussion of related memory-based works.
- For the EgoMem benchmark, they provided a strong rationale for the task design and acknowledged plans to include multi-event interaction questions in future work.

After the discussion, Reviewers RcwM and YuVT are satisfied by the rebuttal and maintained their "Accept" recommendations. Reviewer Dmx9's concerns were also resolved, leading them to maintain a "Borderline accept" score. Reviewer 4wav are still not fully convinced on the points of efficiency. Overall, the authors' thorough rebuttal strengthened the paper and convinced the majority of reviewers towards acceptance. The remaining concerns, while valid, do not undermine the paper's core contributions.